# Non-invasive human skin transcriptome analysis using mRNA in skin surface lipids

Takayoshi Inoue [1✉], Tetsuya Kuwano[1], Yuya Uehara[1], Michiko Yano[1], Naoki Oya[1], Naoto Takada[1], Shodai Tanaka[1], Yui Ueda[1], Akira Hachiya[1], Yoshito Takahashi[1], Noriyasu Ota[1] & Takatoshi Murase [1✉]

Non-invasive acquisition of mRNA data from the skin can be extremely useful for understanding skin physiology and diseases. Inspired by the holocrine process, in which the sebaceous glands secrete cell contents into the sebum, we focused on the possible presence of mRNAs in skin surface lipids (SSLs). We found that measurable levels of human mRNAs exist in SSLs, where the sebum protects them from degradation by RNases. The AmpliSeq transcriptome analysis was modified to measure SSL-RNA levels, and our results revealed that the SSL-RNAs predominantly comprised mRNAs derived from sebaceous glands, the epidermis, and hair follicles. Analysis of SSL-RNAs non-invasively collected from patients with atopic dermatitis revealed increased expression of inflammation-related genes and decreased expression of terminal differentiation-related genes, consistent with the results of previous reports. Further, we found that lipid synthesis-related genes were downregulated in the sebaceous glands of patients with atopic dermatitis. These results indicate that the analysis of SSL-RNAs is a promising strategy to understand the pathophysiology of skin diseases.

[1] Biological Science Research, Kao Corporation, 2606 Akabane, Ichikai-machi, Haga-gun, Tochigi 321-3497, Japan. ✉email: inoue.takayoshi@kao.com; murase.takatoshi@kao.com

Intra- and inter-organ communication mediated via various hormones, growth factors, cytokines, metabolites, and miRNAs play important roles in maintaining homeostasis in the human body[1]. Several efforts have been made to establish comprehensive methods to analyze the expression of these mediators to monitor the physiological conditions of the body and explore predictive biomarkers for various diseases[2–5]. Particularly, the use of serum, urine, and saliva samples, which can be obtained in a non- or low-invasive manner, has been widely investigated.

The skin is often referred to as the window to the body's health since the skin's phenotypes, such as the cutaneous pathology and appearance, and its secretions reflect not only skin conditions but also the conditions inside the body[6]. Moreover, skin forms the body surface and biomolecules can be easily collected from the sweat, hair, and stratum corneum samples, thereby making skin a useful source of samples to monitor skin and body conditions. For instance, the cortisol content in scalp hair correlates with long-term cumulative cortisol exposure[7]. Sweat can also be used as an indicator of internal physiological changes[8], and attempts have been made to use it for monitoring patients' conditions; for instance, the tracking of blood glucose levels by measuring glucose in sweat samples of patients with diabetes[9]. Although metabolites, proteins, and DNA are relatively easy to collect from the sweat and hair samples[10,11], it is difficult to collect measurable levels of mRNAs from skin in a non- or low-invasive manner. So far, tape-stripped stratum corneum has been used to collect mRNAs in a minimally invasive manner;[12] however, the mRNA content thus collected is very low and highly degraded due to RNase activity on the skin surface. Therefore, a skin biopsy is, essentially, required to analyze mRNA expression; however, this method is invasive, which limits its application. More recently, a minimally invasive method for mRNA analysis via RNA-seq using AmpliSeq technology with 16–20 consecutive tape strips has been reported[13–16]. However, tape stripping of the stratum corneum is known to induce skin damage, including disruption of the skin barrier, epidermal hyperproliferation, and infiltration of CD3-positive T cells into the dermis[17], indicating that the problems related to the invasiveness of this technique remain to be fully resolved.

Sebaceous glands synthesize and accumulate lipids to produce sebum. The lipids accumulated in the cytosol of sebocytes are secreted into the sebaceous ducts following the rupture of the plasma membrane. This mode of secretion is called holocrine secretion[18] and is unique to exocrine glands, such as lipid-secreting sebaceous and meibomian glands. The holocrine secretion of cell contents led us to the idea that sebum may contain various biomolecules, including mRNAs, which may be useful for analyzing biological information. Therefore, in this study, we first investigated the presence of mRNAs in human sebum and established a non-invasive, comprehensive method of analyzing human mRNAs using skin surface lipids (SSLs) as samples. Further, the applicability of this method in skin characterization was verified in healthy subjects and in patients with atopic dermatitis (AD).

## Results

**Measurable human mRNA is present in SSLs**. qPCR was conducted to evaluate the mRNA abundance in the SSL samples. The expression of ACTB and GAPDH mRNA in SSLs was comparable that in 100 ng to 500 pg and 1–100 ng of total RNA in normal human epidermal keratinocytes (NHEK), respectively (Fig. 1a). The level of human mRNA degradation in SSLs could not be measured directly due to the presence of bacterial mRNA. Therefore, we established an assay to verify the quality of human mRNA in SSLs. The reverse primer was designed near the 3′ end

and was used with the forward primers to amplify 363 bp and 57 bp long human ACTB mRNAs. After reverse transcription using oligo-dT primers, ACTB levels were quantified by performing qPCR using primers generating 363 and 57 bp human ACTB mRNAs. In the case of ACTB mRNA of longer than 363 bp, both the 363 and 57 bp fragments were amplified, whereas in the case of ACTB mRNA of 57–363 bp, only the 57 bp fragment was amplified (Fig. 1b). This assay was validated using RNA subjected to artificial and gradual degradation using RNase. The gradually degraded RNA was analyzed to calculate the percentage of fragments containing >200 nucleotides (DV200 value) (Fig. 1c, d). The DV200 values of gradually degraded RNA negatively correlated with the abundance ratio of the 363 bp and 57 bp amplicons (Fig. 1e). The level of human mRNA degradation in SSLs collected from six males was calculated using a standard curve (Fig. 1e) and showed a mean DV200 value of 56.5% (Fig. 1f).

**Sebum lipids inhibit RNase activity**. Consistent with the results of a study reporting RNase 7 expression in human skin[19], we confirmed that RNase 7 was expressed in the sebaceous glands and epidermis and also detected in SSLs (Fig. 2a, b). The median RNase 7 content in SSLs collected from 11 subjects was 0.09 ng per mg of sebum (range: 0.026–0.38 ng/mg). Because there is high abundance of RNase on the surface of human skin, it was surprising to detect human mRNA in SSLs. This finding led us to hypothesize that sebum lipids inhibit RNase activity. Based on these results, we evaluated the influence of sebum lipids on recombinant RNase 7 activity. Intact cellular RNA and RNase 7 were incubated with or without sebum lipids obtained from four subjects at 37 °C for 30 min. Interestingly, the 28S and 18S ribosomal RNAs were completely degraded after incubation with RNase 7 in the absence of sebum lipids; on the other hand, ribosomal RNAs were stable in the presence of lipids (Fig. 2c). Next, we aimed to identify the key lipids that inhibit RNase 7 activity. The sebum lipids were separated into fractions A–D on a thin-layer chromatography (TLC) plate (Fig. 2d), and the lipids were recovered from each fraction and subjected to the RNase 7 inhibition assay. We observed that while fractions A, B, and C decreased RNase 7 activity, the inhibitory effect of fraction D was weak (Fig. 2e). Triglycerides and esters in the sebum are hydrolyzed by the skin microbiome to generate free fatty acids (FFAs). The FFAs in human sebum are predominantly composed of 16 carbon atoms (palmitic acid, 16:0; sapienic acid, 16:1Δ6; and palmitoleic acid, C16:1Δ9)[20]. Therefore, we evaluated the inhibitory effects of free palmitoleic acids and various lipids with palmitoleic acid as their main fatty acid, on RNase activity. The free palmitoleic acids and other lipids were dissolved in the reaction buffer at 1 mg/mL or 100 mg/mL concentration, except for cholesterol and wax esters, which could not be dissolved at 100 mg/mL. Our results showed that FFAs strongly suppressed RNase activity at 1 mg/mL concentration, compared with other lipids (Fig. 2f). Moreover, FFAs of different chain lengths (myristoleic acid (C14:1) and oleic acid (C18:1)) also suppressed RNase 7 activity (Fig. 2g).

**Global expression analysis of SSL-RNAs**. For specific and comprehensive quantification of human mRNAs in SSLs, we performed the AmpliSeq transcriptome analysis that can perform multiplexed amplification of cDNA amplicons for more than 20,000 genes. Moreover, this method can analyze even small amounts of RNAs as well as degraded RNAs[21,22]. Although we attempted to prepare sequence libraries from SSL-RNAs based on the default protocol, our success rate was low. Since the data quality can be improved via optimization of the AmpliSeq

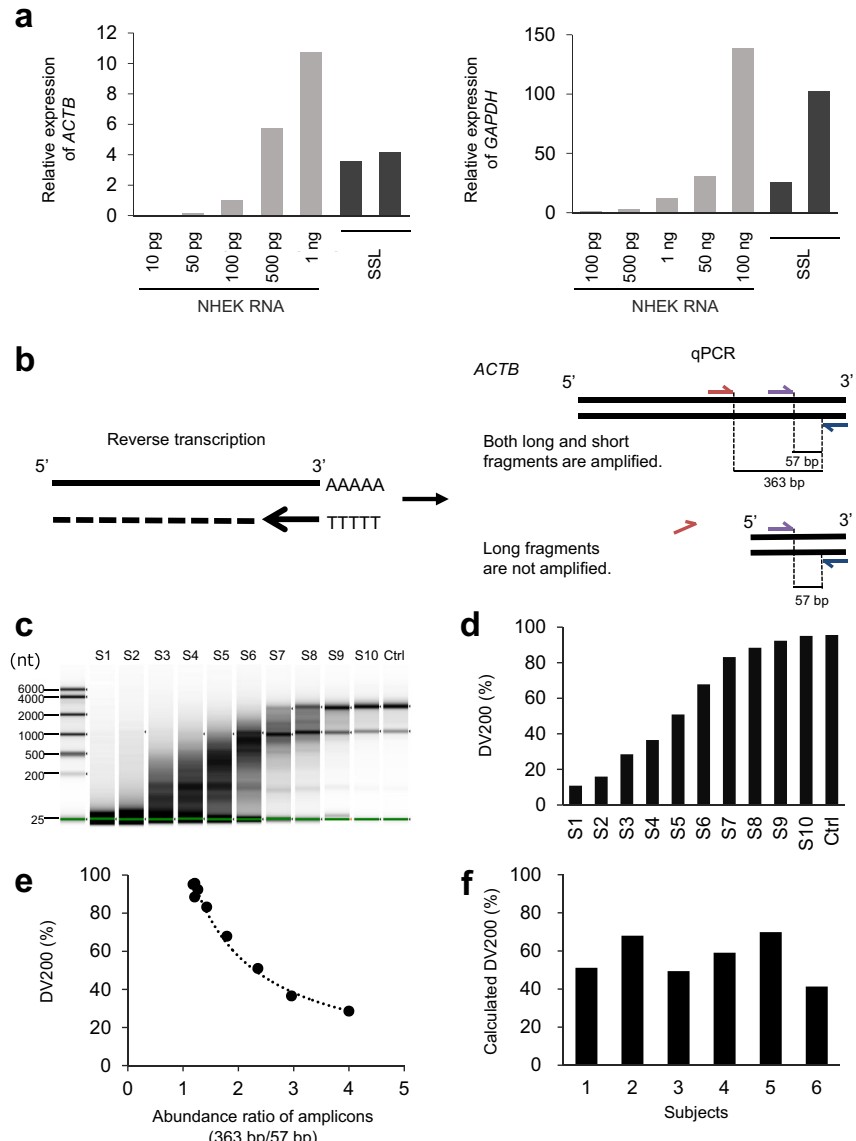

**Fig. 1 Evaluation of mRNA expression and RNA degradation in skin surface lipids (SSLs). a** qPCR to check for *ACTB* and *GAPDH* mRNA expression in SSL samples. Gene expression is shown as the relative expression for 100 pg of RNA derived from normal human epidermal keratinocytes (NHEK). NHEK total RNA (10 pg to 100 ng) is the standard used. **b** Outline for assaying the degradation of human mRNA in SSLs. The extent of mRNA degradation was determined using the long (363 bp)/short (57 bp) amplicon ratio calculated from qPCR results. **c** Preparation of RNA samples with different levels of degradation. A series of standards (S1–S10) are presented with known level of RNA degradation. **d** DV200 value indicating the percentage of fragments containing >200 nucleotides, of **c**. **e** The relationship between the DV200 of **d** and the value of 363 bp/57 bp (abundance of 363 bp amplicon/abundance of 57 bp amplicon measured by qPCR). **f** Indirect assessment of the human mRNA degradation in SSLs, *n* = 6. DV200 of each SSL-RNA sample was calculated using the standard curve of **e**.

protocol[23], we modified the experimental conditions for preparing the sequence library. In the modified protocol, we altered the volume of reagents, standardized the conditions for reverse transcription and target amplification, and added a purification step after the target amplification to remove the primer dimers (Supplementary Method 1). With the new protocol, the success rate of the library preparation improved and the AmpliSeq library was prepared with samples obtained from healthy subjects (91%, 29/32) and patients with AD (100%, 30/30). To analyze the experimental bias of our protocol, we verified the correlation between the expression results of AmpliSeq and qPCR of thymus and activation-regulated chemokine (*TARC/CCL17*) and corneodesmosin (*CDSN*), and observed a high correlation (*CCL17*, R = 0.81, *CDSN*, R = 0.87) (Fig. 3a). Furthermore, the correlation coefficients for the technical replicate of the reverse transcription

(0.94 and 0.90) confirmed that our protocol had low experimental bias (Fig. 3b).

**The SSL-RNA expression profile predominantly reflects mRNA expression in sebaceous glands, epidermis, and hair follicles.** The regions of the sebaceous glands, epidermis, sweat glands, hair follicles, and dermis were isolated from the human skin sections using LMD followed by AmpliSeq transcriptome analysis (Supplementary Fig. 1a). Each region generated distinct clusters when multidimensional scaling was performed to analyze the similarity of the transcriptome profile in different regions (Supplementary Fig. 1b). Next, we focused on the genes highly expressed in each region (Supplementary Fig. 1c)[24–39]. Genes encoding ELOVL fatty acid elongase 3 and 5 (*ELOVL3* and *ELOVL5*), perilipin 2

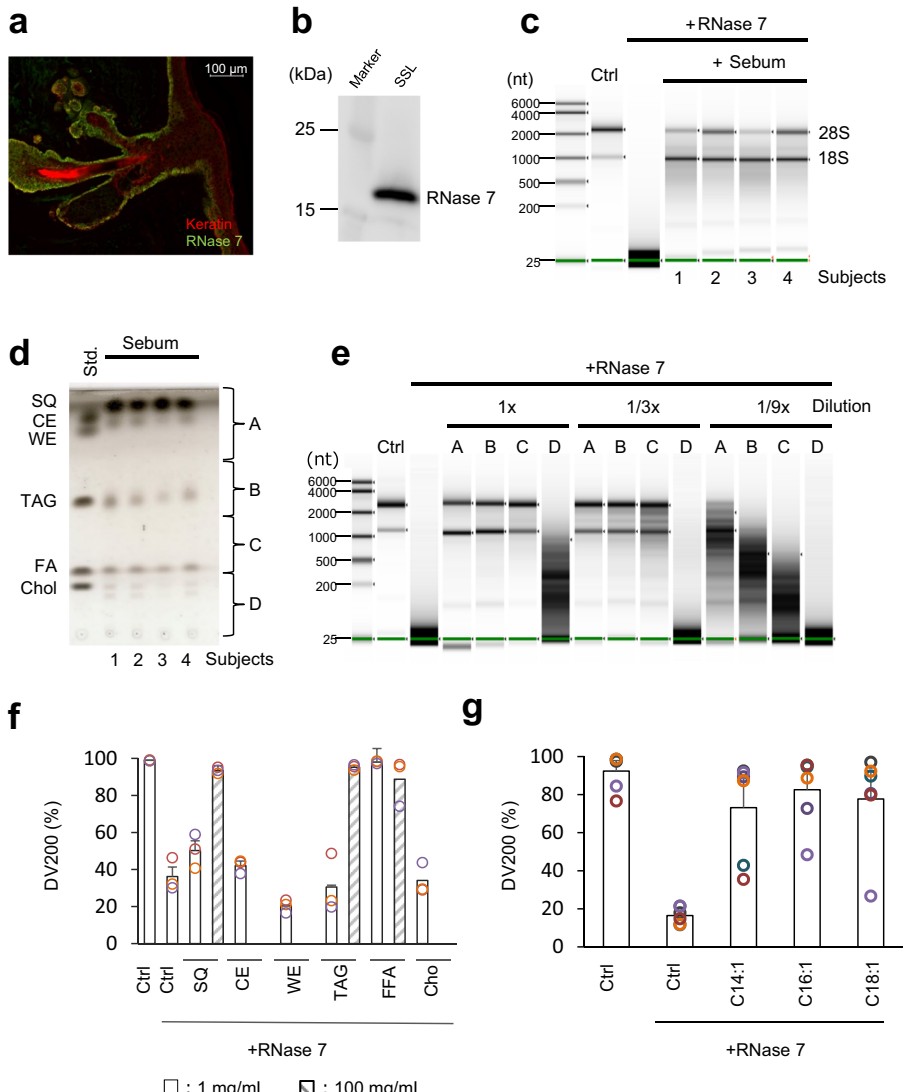

**Fig. 2 Effect of sebum lipids on RNase activity. a** The localization of RNase 7 in the human skin. Green, RNase 7; red, keratin/cytokeratin. Bar: 100 μm. **b** The detection of RNase 7 in SSLs collected from healthy males by western blotting. **c** Determination of the effect of sebum lipids collected from four healthy males on RNase activity using NHEK total RNA. **d** Fractionation of sebum lipids collected from four healthy males by performing thin-layer chromatography (TLC). The standard lane (Std) includes authentic samples: SQ squalene; CE cholesterol ester (cholesteryl palmitate); WE wax ester (lauryl palmitoleate); TAG triacylglycerol (glyceryl trioleate); FA free fatty acid (palmitoleic acid); and Chol cholesterol. A to D sebum samples were collected, and lipids were extracted from the silica gel for the subsequent assays. **e** Effect of pooled sebum lipids collected from A to D on RNase activity using NHEK RNA. **f** Effect of each lipid on RNase activity using NHEK RNA. The DV200 values are shown as the mean with the standard error of the mean (SEM), $n = 3$. WE, wax ester (behenyl palmitoleate); TAG, triacylglycerol (glyceryl tripalmitoleate); Ctrl; EtOH. **g** Relationship between RNase activity and chain length of fatty acids. The DV200 values are shown as the mean with SEM, $n = 6$. C14:1, myristoleic acid; C16:1, palmitoleic acid; C18:1, oleic acid; Ctrl; EtOH.

and 5 (*PLIN2* and *PLIN5*), and microsomal glutathione S-transferase 1 (*MGST1*) are highly expressed in sebaceous glands[24–27]. In our study, these genes were highly expressed in sebaceous glands isolated via LMD than in other regions (Supplementary Fig. 1c). Other regions isolated by LMD also expressed region-characteristic genes, indicating that the LMD was performed successfully.

To gain further insight into the characteristics and origin of SSL-RNAs, we explored genes highly expressed in each region isolated by LMD. In each region, genes with a mean of $\log_2$ (normalized counts + 1) >10, and with more than 1.5-fold differential expression compared to other region(s), were selected (Fig. 4). However, using these criteria, we did not find

any gene selectively expressed in epidermis, due to its similarity with the hair follicles. Therefore, we listed epidermal genes with more than 1.5-fold differential expression than in sebaceous glands, sweat glands, and dermis. To obtain information regarding the origin of SSL-RNAs, the SSL-RNA profile of 29 healthy subjects was analyzed using the genes expressed characteristically at each region. The results presented in the heat map indicate that SSL-RNAs predominantly comprise mRNAs characteristic for sebaceous glands, epidermis, and hair follicles (Fig. 4).

The filaggrin (*FLG*), filaggrin 2 (*FLG2*), and aspartic peptidase retroviral like 1 (*ASPRV1*), that were expressed in the granular layer of epidermis[28–30], were abundantly expressed in SSL-RNAs

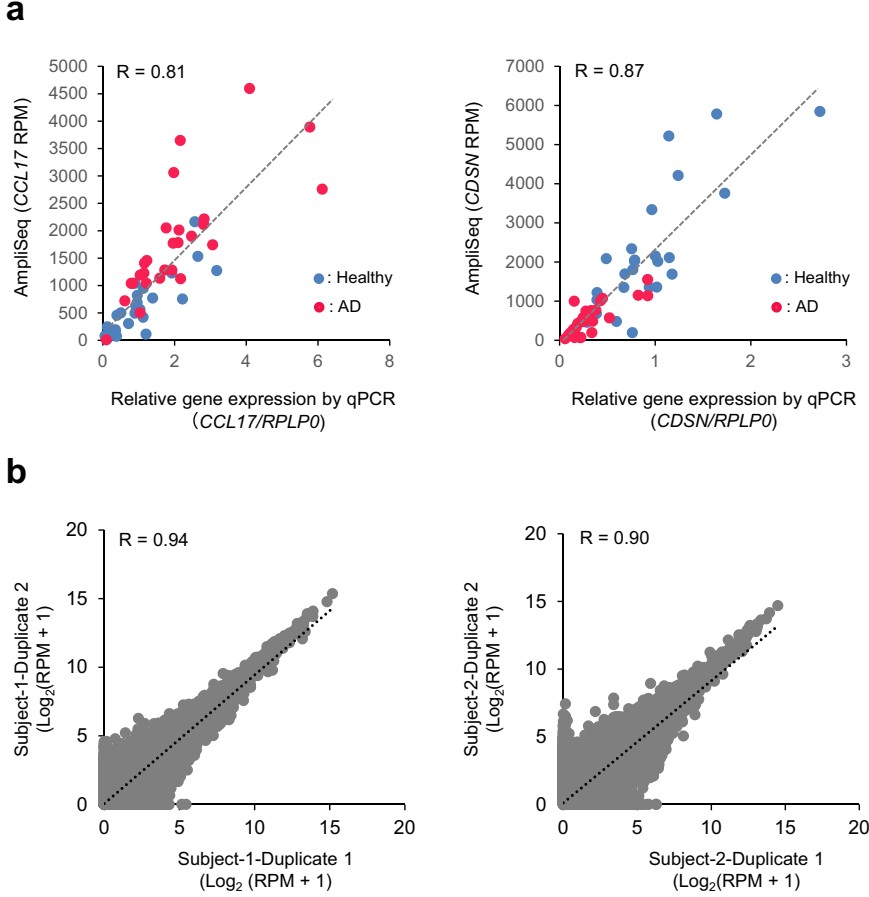

**Fig. 3 Accuracy of the AmpliSeq data output using the modified protocol. a** The correlation between AmpliSeq and qPCR results for *CCL17* and *CDSN* expression in healthy subjects ($n = 29$, blue) and patients with AD ($n = 30$, red). **b** The correlation of the transcriptome profile when preparing libraries in duplicate from each SSL-RNA obtained from two healthy male subjects.

(Fig. 4). The genes encoding keratin 25, 27, and 71 (*KRT25, KRT27,* and *KRT71,* respectively) that are expressed in the inner root sheath of hair follicles[34,35] were highly expressed in SSL-RNAs (Fig. 4). These results suggest that SSL-RNAs provide information regarding the granular layer of the epidermis and the inner root sheath of hair follicles.

To further characterize SSL-RNAs, we compared the transcript profiles of SSLs and the stratum corneum. RNAs present in the stratum corneum were enriched in epidermis-related RNAs, whereas RNAs present in the SSL were also rich in sebaceous gland- and hair follicle-related RNAs (Supplementary Fig. 2). The correlation coefficient between SSL-RNAs and stratum corneum RNAs calculated using transcriptome profile normalized with DESeq2 was 0.51 (the mean value of 10 subjects). When focusing on 34 genes highly expressed in the epidermis, the correlation coefficient increased to 0.82.

The SSL-RNA profiles of the face and scalp, which have dense sebaceous glands and hair follicles, were also compared. The analysis of the dimensionality reduction using t-distributed stochastic neighbor embedding (t-SNE) and variance stabilizing transformation (VST) values in all genes showed that face samples and scalp samples showed to be distinctly classified into two groups (Supplementary Fig. 3). The expression levels of 633 genes were significantly different between the face and scalp, and gene ontology analysis of 305 genes, which were abundantly expressed in the scalp compared with the face, demonstrated the GO terms keratinization (GO:0031424), (FDR = 2.43E−10). Therefore, the expression levels of hair keratins and hair

follicle-specific epithelial keratins[40] were compared between the face and scalp, showing that the expression levels of most keratin molecules were higher in the scalp than in the face (Supplementary Table 1). The expression levels of type 1 hair keratins could not be detected in most of the samples. Thus, the SSL-RNA analysis can be used to characterize skin regions.

Furthermore, we performed a comparative analysis of SSL-RNA profiles in male and female individuals known to have different sebum production levels[41]. The levels of casual sebum on the cheek and the recovered sebum on the forehead and cheek at 60 min after washing the face in male were significantly higher than those in female (Supplementary Fig. 4a). The expression of the SSL-RNAs differed greatly depending on such skin characteristics; the expression levels of 25 lipid metabolism-related genes expressed in sebaceous glands were higher in males, in agreement with recovered sebum levels (Supplementary Fig. 4b).

**Comparison of SSL-RNAs expression between healthy subjects and patients with AD**. We analyzed SSL-RNAs from 29 healthy subjects and 30 patients with AD. The major output of the AmpliSeq data obtained with our modified method was as follows: (i) the average number of reads was 11,456,318 in healthy subjects and 11,137,677 in patients with AD; (ii) the average mapping ratio was 84.0% in healthy subjects and 92.4% in patients with AD; and (iii) the average of the ratio of target detected genes was 44.8% in healthy subjects and 50.2% in patients with AD. First, we verified the expression of genes with significant differential expression in AD. Consistent with previous

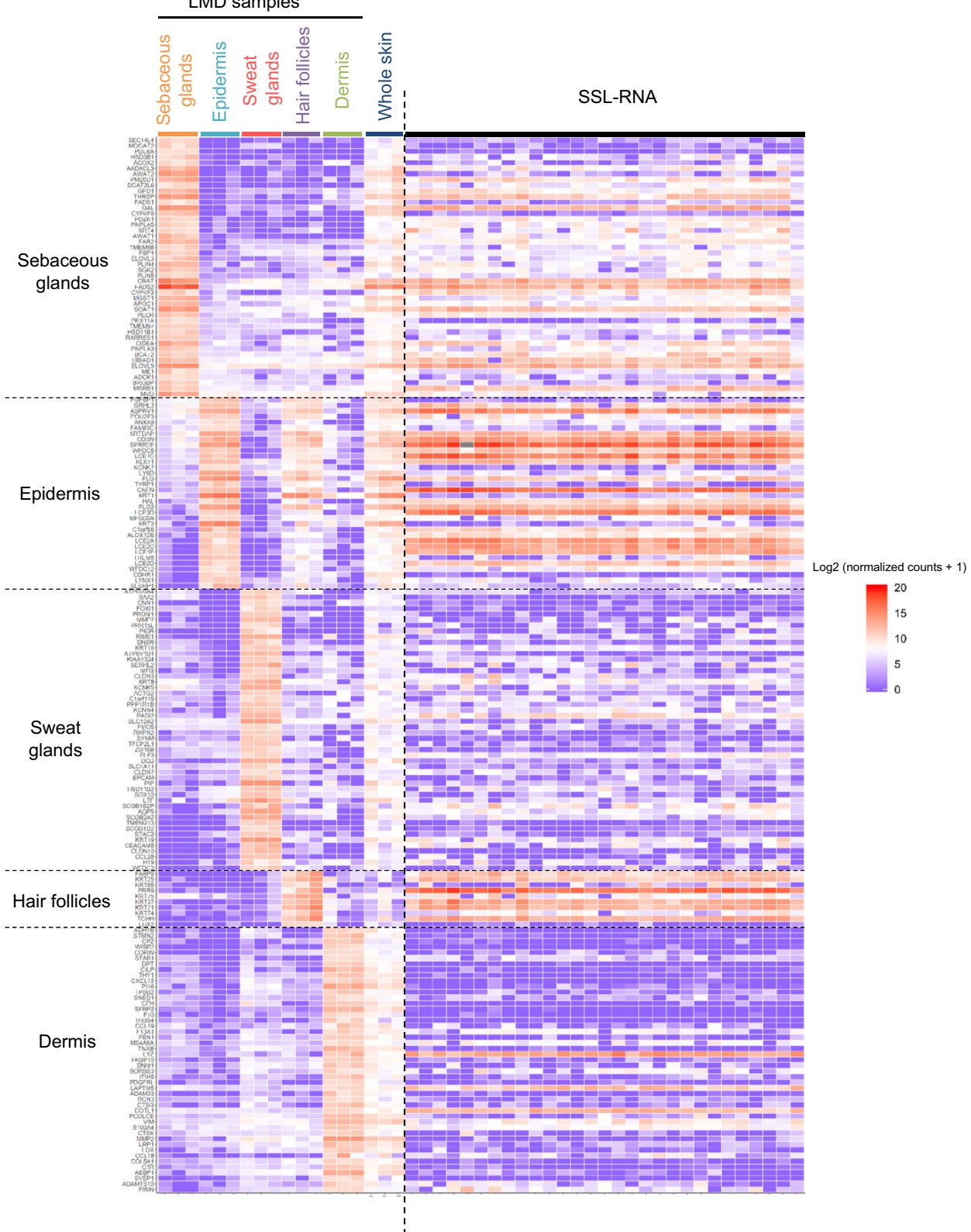

**Fig. 4 mRNA expression characteristic of different skin regions and its comparison with the expression profile of SSL-RNAs.** Heatmap showing the expression profiles for each region (sebaceous glands, epidermis, sweat glands, hair follicles, and dermis) isolated using LMD, whole skin from three healthy male subjects, and SSL-RNAs obtained from 29 healthy male subjects.

reports[42–45], our results in SSL-RNAs analysis showed that the expression of *CCL17*, interleukin 1β (*IL1B*), interleukin 13 (*IL13*), and S100 calcium-binding protein A9 (*S100A9*) significantly increased in patients with AD, but the expression of *FLG* and

involucrin (*IVL*) was significantly decreased in the patients with AD compared to healthy subjects (Fig. 5a). In a previous study reporting a global gene expression analysis in skin biopsy samples, genes related to the terminal differentiation of keratinocytes

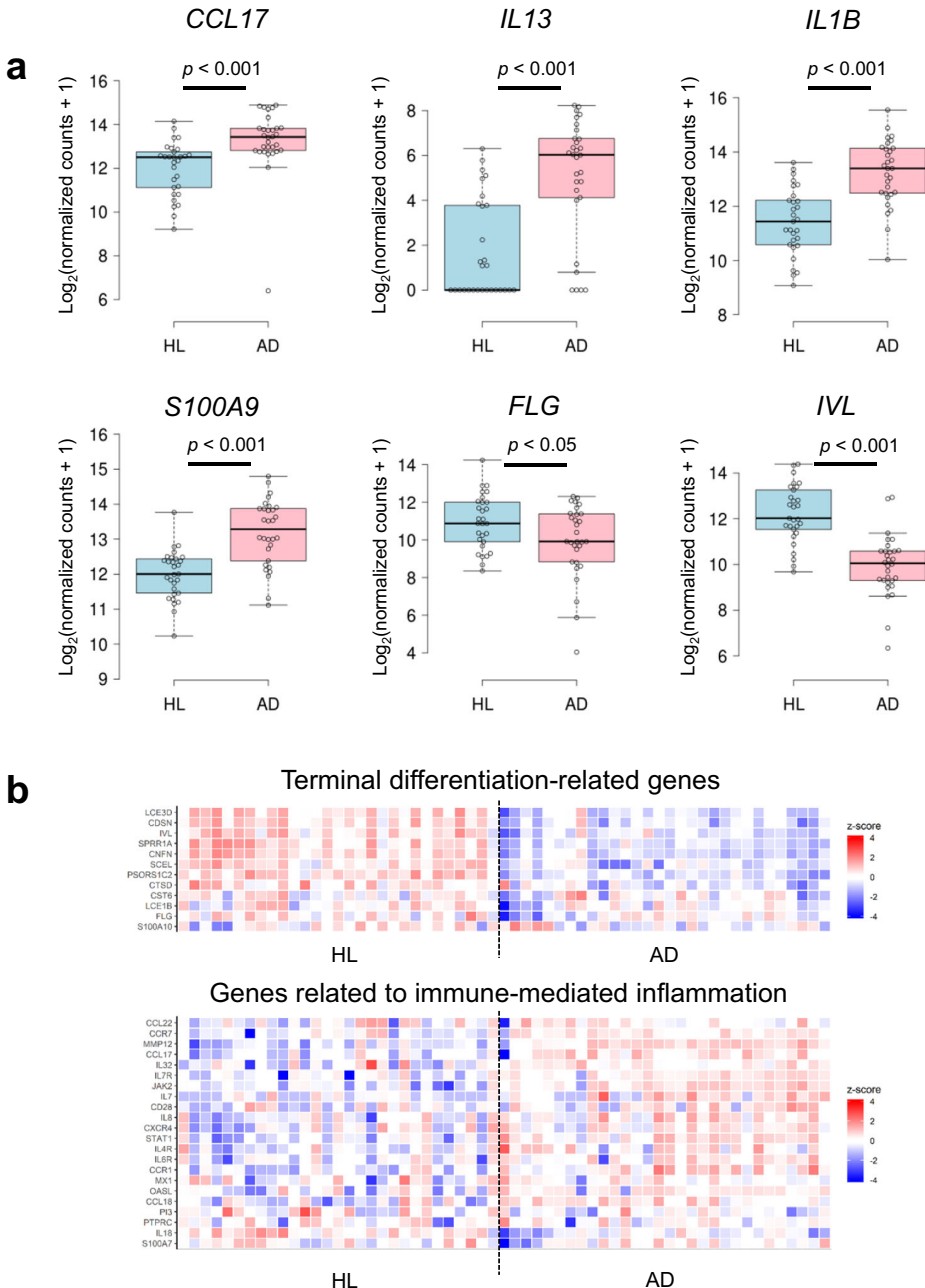

**Fig. 5 Comparison of AD marker genes in healthy subjects (HL) and patients with AD. a** The differential expression of *CCL17*, *IL13*, *IL1B*, *S100A9*, *FLG*, and *IVL* in HL and AD. Boxes represent mean ± interquartile range (IQR), and whiskers represent 1st and 3rd quartile 1.5 * IQR. Benjamini–Hochberg adjusted *p* values are shown from the likelihood ratio test between HL and AD. HL ($n = 29$), AD ($n = 30$). **b** Heat maps using z-transformed log$_2$ (normalized counts + 1) in 12 terminal differentiation-related genes (upper) and 22 genes related to immune-mediated inflammation (lower).

were significantly downregulated and those related to immune-mediated inflammation were upregulated in patients with AD compared with their expression levels in healthy subjects[42]. Based on this report, we selected 12 genes related to terminal differentiation and 22 genes related to immune-mediated inflammation that were detected in SSL-RNAs and compared their expression patterns. Our results showed that the expression patterns of these genes in SSL-RNAs of patients with AD and healthy subjects were largely consistent with the previous report (Fig. 5b).

Moreover, the analysis of the dimensionality reduction using t-SNE and VST values in all genes showed that the healthy subjects and patients with AD could be distinctly classified into two groups (Fig. 6a). To identify the differential biological functions between healthy subjects and patients with AD, we

extracted 918 upregulated and 1033 downregulated genes in patients with AD (Fig. 6b and Supplementary Data 1). For the 918 upregulated genes, the GO terms mRNA splicing and stimulatory C-type lectin receptor signaling pathway were significantly enriched, while the GO term detection of chemical stimulus involved in sensory perception of smell and keratinocyte differentiation was enriched among the 1033 downregulated genes (Fig. 6c).

The sebum secretion is reduced in patients with AD than in healthy individuals[46]; however, the molecular mechanism underlying this reduction remains unknown. GO analysis was performed on 46 genes highly expressed in the sebaceous glands selected from the results of the LMD experiment (Fig. 4), which resulted in the enrichment of genes involved in lipid metabolism

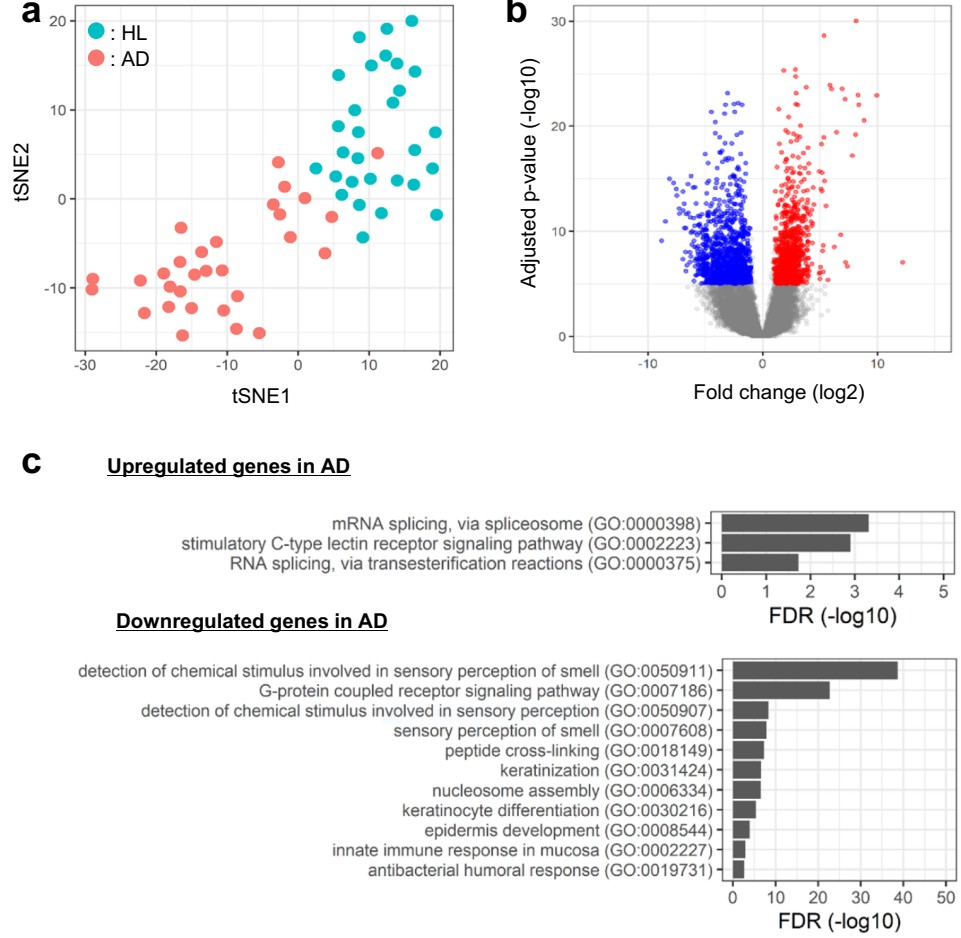

**Fig. 6 Characterization of SSL-RNAs profiles in healthy subjects (HL) and patients with AD. a** t-SNE analysis using variance stabilizing transformation (VST) values for all genes (green, HL; red, AD). **b** Volcano plot of differentially expressed genes (DEGs) (red, upregulated; blue, downregulated) in patients with AD compared to HL subjects (Benjamini-Hochberg adjusted *p* value < $10^{-5}$ and fold change > 2.0). **c** Gene ontology analysis of DEGs. The upper panel shows significant biological process (BP) of upregulated DEGs and the lower panel shows BP of downregulated DEGs in patients with AD (FDR < 0.05).

(Fig. 7a). The expression of 25 genes involved in lipid metabolism (GO:0006629) was downregulated in patients with AD compared to healthy individuals (Fig. 7b). Furthermore, genes encoding peroxisome proliferator-activated receptor alpha (*PPARA*), peroxisome proliferator-activated receptor gamma (*PPARG*), MYC proto-oncogene, bHLH transcription factor (*MYC*), transforming growth factor beta 1 (*TGFB1*), tumor protein p53 (*TP53*), and PR/SET domain 1 (*BLIMP1*) regulate sebocyte differentiation and sebum production in vivo and ex vivo[47–50]. Among these gene, the expression of *TGFB1* was significantly upregulated in patients with AD than in healthy subjects (Fig. 7c).

## Discussion

In this study, we found that mRNAs of measurable quantity and quality were present in SSLs. Further, we established a non-invasive and comprehensive method to profile skin mRNAs using SSLs conveniently collected from the skin surface with an oil-blotting film.

A substantial quantity of RNases is present on the skin surface, and extreme caution should be taken when handling mRNAs. However, unexpectedly, we found that mRNAs in SSLs escape RNase degradation due to the lipid components of the sebum and can be analyzed by AmpliSeq transcriptome sequencing. We analyzed the lipid components responsible for inhibiting the RNase activity and our results suggested that FFAs, triacylglycerol, and squalene contributed to the RNase inhibitory activity of sebum. Due to inhibitory effects of lipids on RNases, we speculate

that mRNAs may be less susceptible to RNases-mediated degradation in the lipid-rich/low-water environment of SSLs. In addition, the optimal pH for RNase activity is 6.5–8.0[51], whereas the skin surface is generally weakly acidic (pH 4.1–5.8) due to the presence of organic acids such as lactic acid;[52] hence, these factors may collectively reduce the RNase activity in SSLs. It is known that sebum is secreted by sebaceous glands in the form of fine granules (4–5 nm)[53]; however, there is no knowledge on the spatial arrangement of lipids, organic acids, and mRNAs in SSLs. Identifying the distribution of these components, as well as the molecular interactions between them, is necessary to understand the precise stabilization mechanism of mRNA on the skin.

We established the method for comprehensive analysis of SSL-RNAs. Recent advances in sequence technology have made it possible to analyze even degraded mRNA. To prepare RNA-seq libraries, a minimum DV200 value of 30% is generally recommended. Using our modified method, the DV200 of human mRNAs in SSLs was approximately 56.5%, making it suitable for transcriptome analysis. However, when the sequence libraries were prepared according to the standard protocol of Ion AmpliSeq Transcriptome Human Gene Expression Kit, the success rate was very low. Use of the optimized protocol (by standardizing conditions of reverse transcription and target amplification, and adding a purification step after target amplification) led to an improvement in library production efficiency, and the transcriptome sequencing resulted in a success rate of 95%. Further, the analytical error of this

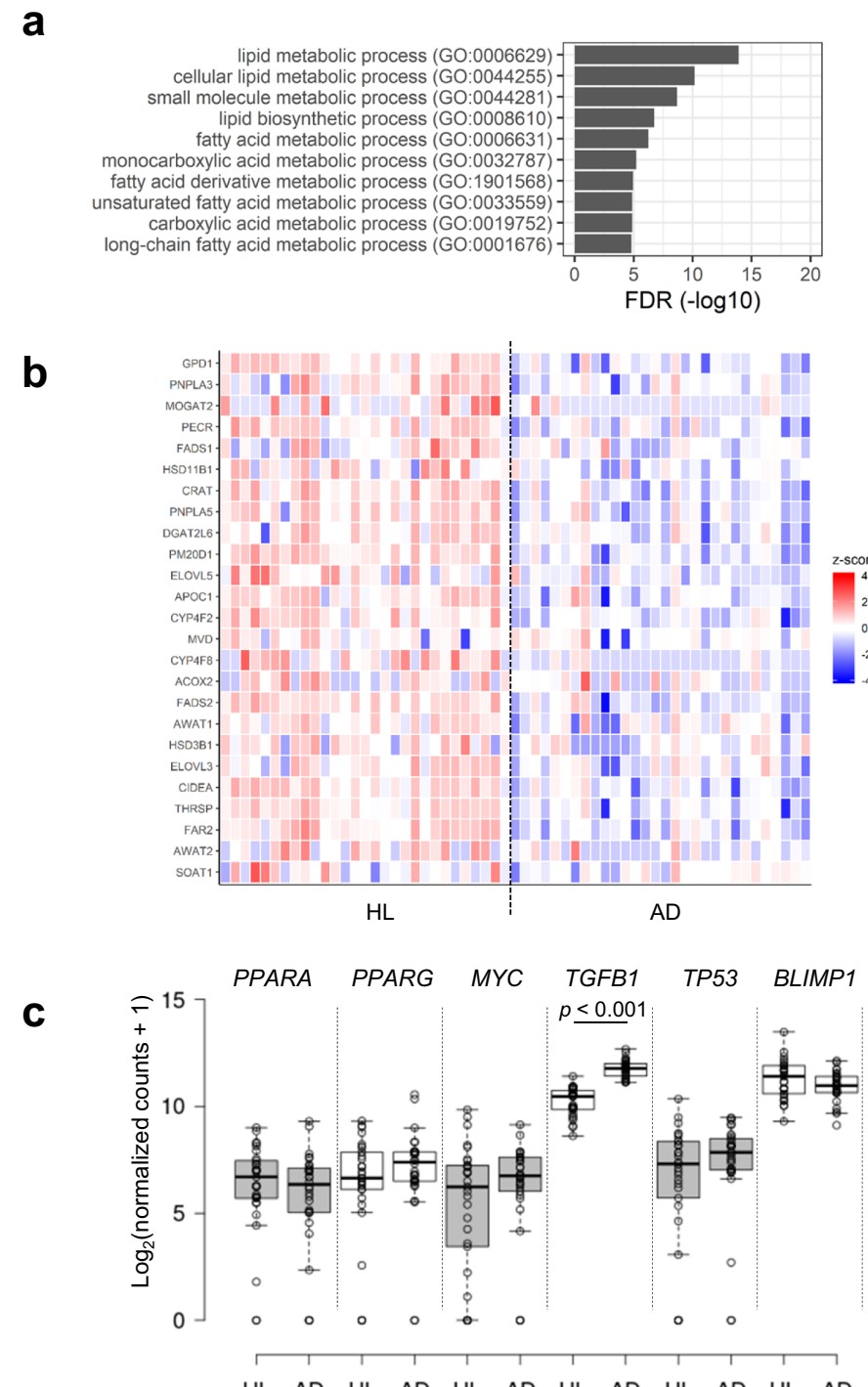

**Fig. 7 Comparison of SSL-RNAs profile representing highly expressed genes in sebaceous glands in healthy subjects (HL) and patients with AD. a** Gene ontology analysis of 25 genes highly expressed in sebaceous glands (selected in Fig. 4). **b** Heat maps using z-transformed log$_2$ (normalized counts + 1) of 25 genes highly expressed in sebaceous glands. **c** The differential expression of *PPARA, PPARG, MYC, TGFB1, TP53,* and *BLIMP1* in HL and patients with AD. Boxes represent mean ± interquartile range (IQR), and whiskers represent 1st and 3rd quartile 1.5 * IQR. Benjamini–Hochberg adjusted *p*-values are shown from the likelihood ratio test between HL and AD. HL ($n = 29$), patients with AD ($n = 30$).

method was low and the results showed high correlation with qPCR results. Thus, our SSL-RNA analysis method using the improved AmpliSeq protocol enables profiling of the mRNA expression in SSLs in a reliable manner.

We investigated the origin of the SSL-RNAs by comparing their mRNA expression profile with those of different regions of the skin. Holocrine secretion of sebum made us speculate that the expression pattern of SSL-RNAs should be similar to that of the sebaceous glands. Indeed, analysis of different regions of the skin obtained by LMD showed that the mRNAs derived from the sebaceous glands were highly expressed in SSL-RNAs. In addition, SSL-RNAs were rich not only in mRNAs derived from sebaceous glands but also in those derived from the epidermis and hair follicles, the tissues in close contact with the sebum. In contrast, the mRNAs characteristic

of sweat glands and dermis, which are not in close contact with sebum, were absent in SSLs. Overall, in contrast to tape-stripped stratum corneum RNAs, SSL-RNAs were thought to characteristically contain RNAs from the epidermis, along with RNAs from the sebaceous glands and surrounding tissues.

The mechanism underlying transfer of epidermal mRNAs into SSLs remain unclear, and there could be several possible mechanisms. The mRNAs characteristic of the granular layer of the epidermis (transcribed from *FLG*, *FLG2*, and *ASPRV1*) were highly expressed in SSL-RNAs. In addition, stratum corneum is reported to contain detectable amounts of mRNA[13–16]. We speculate that the epidermal mRNAs transferred to the stratum corneum due to keratinization are mixed with the sebum on the skin surface, leading to their presence in SSLs. The SSLs also contained hair follicle-derived RNAs, which may be related to the anatomical features of the hair follicle. SSL-RNAs were rich in mRNAs of *KRT25*, *KRT27*, and *KRT71*, the marker genes for the inner root sheath[34,35]. The inner root sheath detaches from the hair shaft and degrades during hair growth, and the process occurs at the orifice of the sebaceous duct[54]. These observations indicate that the epithelial cells of the inner root sheath may get mixed with sebum at the orifice and skin surface, and as a result, information pertaining to the hair follicles may be reflected in the SSL-RNAs. In addition, since extracellular vesicles released from various cells contain several biomolecules including mRNAs[55,56], it is possible that extracellular vesicles-derived mRNAs are also included in the SSL-RNAs. Collectively, our results indicate that SSL-RNAs predominantly contains mRNAs derived from the sebaceous glands, epidermis, and hair follicles, and are, therefore, a useful resource for analyzing the biological information related to the relevant regions of the skin.

Finally, we verified the applicability of this method by performing a comparative analysis of the SSL-RNA profiles of healthy subjects with different amounts of sebum production, and healthy subjects and patients with AD. The SSL-RNA analysis revealed that a group of genes involved in lipid synthesis is highly expressed in sebaceous glands in men with high amounts of recovered sebum, indicating that this method may be useful for detecting skin conditions. Furthermore, we observed that the transcriptome profile was markedly different between healthy subjects and patients with AD, with differential expression of immune-mediated inflammation and terminal differentiation-related genes, as shown in a previous skin biopsy report[42]. Moreover, GO:0050911: detection of chemical stimulus involved in sensory perception of smell was the most significantly changed GO term in our study, consistent with the analysis of stratum corneum RNAs collected from patients with AD[13], indicating that the analysis of SSL-RNAs successfully captured the characteristics of AD. Atrophy of the sebaceous glands and reduction in sebum secretion have been reported in patients with AD[57]. However, little is known about the underlying mechanism, including the gene expression profile of sebaceous glands, in patients with AD due to difficulty in obtaining facial skin tissue samples containing sebaceous glands. Here, we showed that the expression of 25 lipid metabolism-related genes highly expressed in sebaceous glands were lower in AD patients than in healthy subjects. Moreover, the expression of *TGFB1*, which suppresses sebocyte differentiation and lipid accumulation[50], was significantly increased in SSL-RNAs of patients with AD. Increase in the number of TGF-β-positive cells in patients with AD has also been reported[58]. These results suggest that the suppression of lipid synthesis via *TGFB1* may be one of the mechanisms responsible for dysregulated sebum synthesis in these patients. The possibility of TGF-β-mediated regulation of sebum production requires further investigation. Thus, the transcriptome analysis of SSL-RNAs can evaluate the molecular profile of AD in a non-invasive manner, and is a promising method for comprehensive understanding of AD pathology.

There is one limitation regarding the clinical findings of the SSL-RNA analysis. In this study, we validated this method in a relatively limited number of patients (samples collected from 29 healthy male subjects and 30 patients with AD aged 20–48 years were used to validate this method). Validation in a more subjects and on a larger scale is needed to further strengthen the findings of this study and to confirm the applicability of the SSL-RNA analysis method.

In summary, we established a non-invasive method for skin RNA analysis that utilizes SSL samples collected by simply wiping the skin surface for less than a minute. This non-invasive method has potential application in unraveling the molecular profiles of skin conditions and diseases including AD, which will be helpful for clinical management of these diseases in the future. Understanding the status and course of AD at the molecular level is essential not only to assess the pathophysiology of AD but also to design its effective therapeutic treatments. The clinical phenotypes of AD are extremely complex, warranting the need for identifying biomarkers that can classify invisible endophenotypes[59]. This comprehensive analysis method using SSL-RNAs is expected to contribute to solving these clinical and research issues owing to advantageous features such as non-invasiveness and ease of sampling. Further, since the skin is called the disease-sensor organ and is believed to reflect the conditions inside the body[60,61], the SSL-RNA analysis may have wide applicability to understand various pathologies of human body.

## Methods

**Subject recruitment and collection of SSLs.** For SSL-RNA analysis in healthy subjects and in patients with AD, participants were recruited as follows. Thirty-two healthy male individuals (mean age: 34.6 years, range: 20–49 years, SD = 9.24) were recruited in October 2016. The participants were evaluated by dermatologists prior to the commencement of the study to confirm that no obvious skin disease or conditions were present on their faces. Subjects with self-reported AD were recruited, and 30 male patients with mild and moderate facial AD (mean age: 31.0 years, range: 20–48 years, SD = 8.82) diagnosed by the dermatologist between June and October 2017 were then included in the study. To compare SSL-RNAs and stratum corneum RNAs, 10 healthy male individuals (mean age: 40.6 years, range: 28–57 years, SD = 10.23) were recruited. Furthermore, to evaluate the differences of SSL-RNA profile between face and scalp, 10 healthy female individuals (mean age: 26.3 years, range: 24–32 years, SD = 2.41) were recruited. To analyze the relationship between sebum secretion and SSL-RNA profile, 38 healthy male individuals (mean age: 39.1 years, range: 20–58 years, SD = 12.5) and 42 healthy female individual (mean age: 38.5 years, range: 20–56 years, SD = 11.6) were recruited. The participants received an adequate explanation of the study and provided written informed consent. All subjects were told not to remove facial sebum by washing, using wipes, or shaving their face from the test day until the end of the test. Patients with AD were also restricted from using steroidal anti-inflammatory and immunosuppressive drugs on their facial skin from 24 h prior to the study. This study was approved by the Human Research Ethics Committee, Kao Corporation (approval numbers: 792-2016082, T003-170413, T173-180920, and T246-190815), the Japan Aesthetic Dermatology Symposium (approval number: KU-2017-05-003), and the Shinjukuminamiguchi Dermatologic Clinic (approval number: KU-2016-10-005).

SSLs were collected by wiping the whole face (forehead, cheek, face line, nose, and chin) or parietal scalp using an oil-blotting film (8.0 cm × 5.0 cm, 3 M Japan, Tokyo, Japan) and samples were stored in glass vials at −80 °C until use.

To compare SSL-RNAs and stratum corneum RNAs, SSLs were collected according to the aforementioned method. Then, SSLs were completely removed from the face using facial cleaners, and five stratum corneum samples were collected from each left and right cheek using D-square tape strips (22 mm in diameter, CuDerm, Dallas, TX, USA). Samples were stored at −80 °C until use.

**Skin tissue.** Surgically removed adult forehead skin from three Caucasian males, aged 62, 66, and 67 years, and nose skin from one adult (Caucasian female, 66-year-old) were provided by the Colorado Dermatology Institute (Colorado Springs, CO, USA) for laser microdissection (LMD) and immunostaining, respectively. The procurement of skin tissues was approved by the Institutional Review Board of IntegReview Ltd. (Austin, TX, USA; approval number: T046a-170829) and was conducted according to the Declaration of Helsinki Principles. Informed consent was obtained from the volunteers prior to surgery. After surgery, the skin tissues were stored in

William's E medium (Life Technologies, Carlsbad, CA, USA) at 4 °C until embedding. All skin tissues were embedded using the Tissue-Tek optimal cutting temperature compound (Sakura Finetek, Tokyo, Japan) and kept frozen until sectioning.

**Measurement of sebum secretion**. Facial sebum levels on the forehead and cheek were measured using a Sebumeter (SM815, C-K Electronics, Cologne, Germany). The casual sebum levels before washing the face and the recovered sebum levels during 60 min after washing the face were measured[62].

**mRNA extraction and qPCR**. RNAs in SSLs (SSL-RNAs) were extracted using the TRIzol reagent (Thermo Fisher Scientific, Waltham, MA, USA) as follows: 2.85 mL of TRIzol was added to a finely cut oil-blotting film containing sebum samples. Next, the solution was divided equally into two tubes and 260 μL of chloroform was added to each tube and mixed by vortexing. The tubes were centrifuged at 12,000 × g for 15 min at 4 °C. The upper layer was transferred to a fresh tube and precipitated with ethanol. The precipitates were washed with 70% ethanol (v/v) and dissolved in 10 μL of nuclease-free water.

Reverse transcription was performed using the SuperScript IV First-Strand Synthesis System and Oligo-dT primers (Thermo Fisher Scientific). qPCR was performed using the TaqMan Fast Universal PCR Master Mix (Thermo Fisher Scientific) and TaqMan probes for each gene (Thermo Fisher Scientific).

**Evaluation of mRNA degradation in SSLs**. To prepare standard samples with different levels of degraded mRNAs, total RNA (1 μg) extracted from NHEK (Cascade Biologics, Portland, OR, USA) was incubated with 30–1000 ng/mL of recombinant human RNase 7 (Novus Biologicals, Littleton, CO, USA) in 10 mM Tris-HCl buffer (pH 8.0) (Nippon Gene, Tokyo, Japan) for 30 min. QIAzol lysis reagent (Qiagen, Hilden, Germany) and chloroform were added to the samples treated with RNase 7 or SSLs collected from six healthy males as described above. After the solutions were vortexed and centrifuged, RNA in the supernatant was purified using the miRNeasy Mini Kit (Qiagen). The level of RNA degradation in the samples was determined using the High Sensitivity RNA ScreenTape (Agilent Technologies, Palo Alto, CA, USA) on an Agilent 4200 TapeStation system (Agilent Technologies). The level of degradation of human mRNAs in SSLs was estimated using the following calculated DV200 value, which evaluates the percentage of fragments containing >200 nucleotides. mRNA was reverse-transcribed using the SuperScript IV First-Strand Synthesis System and Oligo-dT primers (Thermo Fisher Scientific), and was amplified in a thermal cycler using the PowerUP SYBR Green Master Mix (Thermo Fisher Scientific) and the following primers:

ACTB (forward primer 57 bp): 5′-GCTTTTGGTCTCCCTGGGAG-3′
ACTB (forward primer 363 bp): 5′-ACAATGTGGCCGAGGACTTT-3′
ACTB (reverse primer): 5′-AGTCAGTGTACAGGTAAGCCC-3′.

Abundance of each amplicon was determined from the RNA standard curve. Further, DV200 values of the SSL-RNA samples were calculated from the standard curve of DV200 plotted against abundance ratio of the amplicons (363 bp/57 bp).

**Immunostaining**. Skin sections obtained from the nose of a Caucasian female were fixed in acetone at −20 °C for 10 min and treated with 0.1% Triton X-100/PBS for 5 min. The skin sections were incubated with Protein Block serum-free (Agilent Technologies) for 30 min, then with the primary antibodies against keratin/cytokeratin (mouse monoclonal (AE-1/AE-3), original solution, Nichirei, Tokyo, Japan) or RNase 7 (rabbit polyclonal, 1:50, Cloud-Clone Corp., Houston, TX, USA) for 1 h at 20–25 °C, and finally with anti-rabbit IgG (donkey polyclonal, Alexa Fluor 555, 1:1000, Thermo Fisher Scientific) or anti-mouse IgG (goat polyclonal, Alexa Fluor 647, 1:1000, Thermo Fisher Scientific) for 30 min. Samples were mounted on a glass slide and imaged using fluorescence microscopy (BZ-X710, Keyence, Osaka, Japan).

**Western blotting**. Total protein was extracted from SSLs using 350 μL of RIPA buffer and was purified using the Ready Prep 2-D cleanup kit (Bio-Rad, Hercules, CA, USA); protein concentration was measured using the BCA protein assay kit (Thermo Fisher Scientific). Afterward, 70 μL of trichloroacetic acid was added and the samples were incubated on ice for 30 min, followed by centrifugation at 13,000 × g for 5 min at 4 °C. Chilled acetone (500 μL) was added to the pellets, the tubes were centrifuged at 13,000 × g for 5 min at 4 °C, and the supernatant was removed. The pellets were dissolved in SDS sample buffer (Novagen, Darmstadt, Germany) and boiled at 95 °C for 5 min. Proteins (5 μg) were separated on a 4–15% polyacrylamide gradient gel (Bio-Rad) and then transferred to a polyvinylidene fluoride (PVDF) membrane (Bio-Rad) soaked in Tris-Glycine Buffer (25 mM Tris, 192 mM glycine; pH 8.2) containing 20% (v/v) methanol. The PVDF membrane blocked with PVDF Blocking Reagent for Can Get Signal (Toyobo, Tokyo, Japan) was incubated with anti-RNase 7 antibody (rabbit polyclonal, 1:200, Cloud-Clone Corp, Katy, TX, USA) for 60 min followed by incubation with anti-rabbit IgG, horseradish peroxidase-linked secondary antibody (donkey monoclonal, 1:2000, GE Healthcare, Bucks, UK) for 45 min. The bands were visualized using ECL prime western blotting detection reagents (GE Healthcare).

**Enzyme-linked immunosorbent assay for RNase 7**. After the facial sebum collected from 11 healthy subjects was weighed, total protein was extracted using

400 μL of RIPA buffer supplemented with protease/phosphatase inhibitor (Cell Signaling Technology, Danvers, MA, USA). The concentration of RNase 7 was measured using an RNase 7 ELISA kit (Hycult Biotech, Uden, Netherlands) according to the manufacturer's protocol.

**Effect of sebum lipids on RNase activity**. The oil-blotting film used to collect SSLs from the face was finely cut, and 1 mL distilled water and 4 mL tert-butyl methyl ether were added to the cut film. The solution was transferred to a new glass vial and centrifuged at 2,050 × g at 4 °C for 10 min. The upper layer was transferred to a new glass vial and the organic solvent was dried by blowing nitrogen over it. The remaining sebum lipids were dissolved in 100 μL of dimethyl sulfoxide (DMSO) and used for analysis.

Cholesterol ester (cholesteryl palmitate, Sigma-Aldrich, St. Louis, MO, USA, C6072), wax ester (lauryl palmitoleate, Santa Cruz Biotechnology, Santa Cruz, CA, USA, sc-280908), triacylglycerol (glyceryl trioleate, Sigma-Aldrich, T7140), free fatty acid (palmitoleic acid, Sigma-Aldrich, P9417), squalene (Sigma-Aldrich, S3626), and cholesterol (Sigma-Aldrich, C8667) were used as authentic samples to identify the lipid molecular species. The sebum lipids extracted from the oil-blotting films were dissolved in chloroform/methanol (2:1, v/v). The authentic lipids and 5 mg sebum lipid samples were separated on a TLC plate using hexane: diethyl ether: acetic acid (70:30:1, v/v/v). After chromatography, the plate was divided using a glass cutter into portions containing the authentic lipids and sebum lipid samples, and the portion containing the authentic lipids was sprayed with a solution containing 10% (w/v) copper sulfate and 8% (w/v) phosphoric acid, and then heated at 180 °C for 3 min to visualize the location of each lipid. Based on the mobility of the authentic lipids, the silica corresponding to portions of sebum lipids was scraped off. The silica was sonicated in the chloroform/methanol (2:1, v/v) mixture to extract the lipids and centrifuged at 2,050 × g for 5 min. The supernatant was dried and dissolved in 15 μL of DMSO; the solution was further diluted threefold and ninefold using DMSO.

Cholesterol ester (cholesteryl palmitoleate, Olbracht Serdary Research Laboratories, Toronto, Canada, D-161), wax ester (behenyl palmitoleate, Nu Chek Prep, Elysian, MN, USA, WE-1368), triacylglycerol (glyceryl tripalmitoleate, Sigma-Aldrich, T5888; glyceryl trioleate, Sigma-Aldrich, T7140), free fatty acids (myristoleic acid, Sigma-Aldrich, M3525; palmitoleic acid, Sigma-Aldrich, P9417; oleic acid, Sigma-Aldrich, O1008), squalene (Sigma-Aldrich, S3626), and cholesterol (Sigma-Aldrich, C8667) were used to determine the active ingredients in the sebum. NHEK RNA (20 μg/mL final concentration) was mixed with RNase 7 (final concentration: 1 μg/mL) in 10 mM Tris-HCl buffer (pH 8.0) with or without sebum or the lipid reagent (total 50 μL), and sonicated, followed by incubation at 20–25 °C for 30 min. Subsequently, RNA was extracted using the TRIzol LS reagent, and the quality of the RNA was determined using the High Sensitivity RNA ScreenTape on the Agilent 4200 TapeStation System.

**Library preparation for Ion AmpliSeq**. Following the addition of 2.85 mL of QIAzol reagent (Qiagen) to a finely cut oil-blotting film containing sebum samples, QIAzol solution was divided equally into two tubes. Chloroform (260 μL) was added to each tube and vortexed, and the tubes were centrifuged at 12,000 × g for 15 min at 4 °C. The upper layer was transferred to a fresh tube. RNA was purified using the RNeasy mini kit (performing DNase treatment in the purification step) (Qiagen) and eluted from the resin twice using 50 μL of nuclease-free water followed by ethanol precipitation, and then dissolved in 10 μL of nuclease-free water.

To improve the success rate of the library preparation using the AmpliSeq protocol, we modified the default protocol of the Ion AmpliSeq Transcriptome Human Gene Expression kit (Thermo Fisher Scientific). Briefly, 1.75 μL of RNA solution was mixed with 0.5 μL of VILO Reaction Mix and 0.25 μL of SuperScript III Enzyme. Reverse transcription was performed at 25 °C for 10 min, 42 °C for 90 min, and finally at 85 °C for 5 min. Target DNA amplification was performed by mixing 2.5 μL of cDNA solution, 1.5 μL of nuclease-free water, 2.0 μL of the Ion AmpliSeq HiFi Mix, and 4.0 μL of the Ion AmpliSeq Transcriptome Human Gene Expression Core Panel under the following conditions: 99 °C for 15 s and 62 °C for 16 min for 20 cycles. The amplified DNA library was purified by mixing 10 μL of AMPure XP beads (Beckman Coulter, Miami, FL, USA) according to the manufacturer's protocol and eluted using 10 μL of nuclease-free water. The quality check of the DNA library was conducted using the High Sensitivity D1000 ScreenTape on the Agilent 4200 TapeStation. When the DNA library had amplified, a band of approximately 170 bp was observed. After checking the DNA library quality, the reaction solution was prepared by mixing 3.5 μL of purified library solution, 2.0 μL of Ion AmpliSeq HiFi Mix, 4.0 μL of Ion AmpliSeq Transcriptome Human Gene Expression Core Panel, and 0.5 μL of VILO Reaction Mix. After adding 1.0 μL of FuPa reagent to 10 μL of reconstituted reaction solution, the primer sequence was partially digested under the following conditions: 50 °C for 10 min, 55 °C for 10 min, and 60 °C for 20 min. To ligate the adapter sequence, 2 μL of Switch solution, 1 μL of Ion Xpress Barcode adapters, and 1 μL of DNA ligase were added to 11 μL of the reaction solution, followed by incubation at 22 °C for 60 min and 72 °C for 5 min. The library (15 μL) ligated with the adapter sequence was purified via mixing with 18 μL of AMPure XP beads according to the manufacturer's protocol. Libraries were eluted using 50 μL of Library Amp Mix (Thermo Fisher Scientific) to which 2 μL of the Library Amp Primers were added. The library amplification was conducted using the following

steps: 98 °C for 15 s and 64 °C for 1 min for five cycles. Next, 50 μL of the PCR product was mixed with 25 μL of the AMPure XP beads and the supernatant was transferred to fresh PCR tubes. The supernatants were mixed with 60 μL of the AMPure XP beads and purified; target fragments were eluted from the beads using 10 μL of TE buffer. The quality check of the library was performed using the High Sensitivity D1000 ScreenTape on the Agilent 4200 TapeStation. We have described the library preparation for AmpliSeq in the comparison between SSL-RNAs and stratum corneum RNAs in Supplementary Method 2.

**Sequencing**. The library was quantified using the Ion Library TaqManTM Quantitation Kit (Thermo Fisher Scientific). After an input of 50 pM of DNA library in the Ion Chef System (Thermo Fisher Scientific), template preparation and chip loading were performed, and RNA-seq was conducted on the Ion S5 XL System (Thermo Fisher Scientific).

**Verification of the correlation coefficient between AmpliSeq and qPCR results**. For identifying *RPLP0*, *CDSN*, and *CCL17* expression in SSL-RNAs, cDNA was pre-amplified with 14 cycles using the TaqMan PreAmp Master Mix (Thermo Fisher Scientific) and the pooled TaqMan probe (*RPLP0, CDSN*, and *CCL17*) (Thermo Fisher Scientific), and then diluted fivefold with nuclease-free water. qPCR was performed as described in the mRNA extraction and qPCR sub-section. Expression value of *RPLP0* was used as an internal control. Correlation between the value of reads per million mapped reads of AmpliSeq and the relative expression value of qPCR in healthy subjects and patients with AD was analyzed.

**Laser microdissection (LMD) and AmpliSeq transcriptome analysis**. Frozen skin sections (thickness: 10 μm) from three Caucasian males were mounted on membrane slides (PEN-Membrane 2.0 μm, Leica Microsystems, Wetzlar, Germany) treated with 0.1% (w/v) poly-L-lysine (Fujifilm Wako Pure Chemical, Osaka, Japan). In addition, two frozen sections were directly collected into 750 μL of RLT buffer (Qiagen) containing 40 mM dithiothreitol (Sigma-Aldrich) to analyze the transcriptome of the whole tissue. The LMD sections were fixed with acetone at −20 °C for 10 min, stained with 0.05% (v/v) toluidine blue, and finally dried. The epidermis, sebaceous glands, sweat glands, hair follicles, and dermis were carefully microdissected from the skin sections using LMD7000 (Leica Microsystems). Fifteen target regions from three skin tissues were dissected and dissolved in 50 μL of RLT buffer containing 40 mM dithiothreitol. Total RNA was extracted and purified using the RNeasy mini kit (performing DNase treatment in the purification step) (Qiagen). The RNA was concentrated via ethanol precipitation and dissolved in 10 μL of nuclease-free water. The RNA quality was checked on the 4200 TapeStation System and a cDNA library was prepared for the AmpliSeq transcriptome sequencing using 75 pg of total RNA according to the method described in the Library preparation for Ion AmpliSeq sub-section. The number of cycles required to achieve target amplification was changed from 20 to 18 cycles, except for in the dermis samples.

**Normalization and differential expression analysis of the AmpliSeq whole transcriptome**. The AmpliSeq RNA plugin of the Ion Torrent Suite Software Plugins (Thermo Fisher Scientific) was used for the primary analysis of the sequencing data. All statistical analyses and normalization of the RNA-seq transcriptome data were performed using the R statistical language. Read counts were generated using the AmpliSeq RNA plugin in the Ion Torrent Suite Software (Thermo Fisher Scientific) and normalized with the DESeq2 R package (Bioconductor). For differential expression analysis between healthy subjects and patients with AD, a likelihood ratio test was performed with DESeq2 using normalized counts. Heat maps were generated using the heatmap3 package. Dimensionality reduction using t-SNE was performed using the Rtsne function of the Rtsne package. All plots were generated using the tidyverse package in combination with the reshape2, gplots, ggplot2, grid, and cowplot packages.

**Statistics and reproducibility**. Unpaired Student's *t*-tests (two-tailed) were used for comparing casual and recovered sebum from the skin samples of male and female. $p < 0.05$ was considered to indicate significant differences. The values shown in Fig. 2f, g represent the mean with the standard error of the mean. The box plots shown in Figs. 5a, 7c, and Supplementary Fig. 4a represent the mean ± interquartile range, and whiskers represent 1st and 3rd quartile 1.5 * IQR.

**Reporting summary**. Further information on research design is available in the Nature Research Reporting Summary linked to this article.

## Data availability

RNA-seq data from healthy subjects and patients with AD (Figs. 3–7), from SSL-RNAs and stratum corneum (Supplementary Fig. 2), and from face and scalp (Supplementary Fig. 3 and Supplementary Table 1) have been deposited in the Japanese Genotype-phenotype Archive (JGA), under the accession code JGAS000416, JGAS000417, JGAS000418. The data that support the findings of this study are available from the corresponding author upon reasonable request. The full blot images for all western blot analyses are included in Supplementary Fig. 5. Source data is available as Supplementary Data 2.

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

## Author contributions

T.I. conceived the study. T.I., A.H., Y.T., and T.M. planned the study. T.I., T.K., Y.U., M.Y., N.T., S.T., Y.U., and N.Oy. performed the experiments and analyzed the data. T.M., Y.T., and N.Ot. supervised the research. T.I. and T.M. wrote the manuscript. All authors reviewed the manuscript.

## Competing interests

A patent application related to this work has been filed (No. PCT/JP2017/021040: method for preparing nucleic acid sample). Status: patent granted (DE, FR, GB, KR, CN, JP), patent pending (US). Inventors: T.I. and A.H. Patent applicant: Kao Corporation). All other co-authors declare that they have no competing interests.
