## [Peer Review File · Communications Biology]

Reviewers' comments:

Reviewer #1 (Remarks to the Author):

Inoue et al. describe a new non-invasive method for sampling and analysing RNA from human skin. The development of new methods for this purpose is key to a better understanding of the skin biology in inflammatory skin disease.

The study samples skin surface RNA (so-called skin surface lipid RNA) by the use of an oil-blotting film to collect facial sebum and isolates RNA from this film by extraction. The RNA is then analysed by qPCR and RNA Sequencing.

It is shown that human mRNA is sensitive to RNase 7, that RNase 7 is present in human skin and that human sebum lipids inhibit RNase 7 mediated breakdown of human mRNA.

Analysis of the transcriptomic profile of different skin regions isolated by laser micro-dissection and comparison to the lipid associated RNA suggest that the RNA is derived from sebaceous glands, epidermal cells and hair follicles, but not sweat glands and dermis.

Comparison of gene expression profiles show that both immune-related and terminal differentiation genes are differentially regulated between healthy subjects and atopic dermatitis patients.

COMMENTS

1. The use of an oil-blotting film and extraction by QIASol indicate that the resulting RNAs are associated with lipids. The transcriptomic analysis further supports that they are derived from sebaceous glands.

It is, however, possible that the isolated skin surface RNA could also be derived from other parts of the skin and is not necessarily associated with lipids from the sebaceous glands. A control experiment with a different isolation material (skin wipe) and extraction buffer could be relevant.

The high level of RNA isolated with this method indicates that the RNAs are protected by lipids. However, the RNase 7 experiments only provide indirect evidence. It is true that RNase 7 exists in the skin but it is not clear to what extent it protects the surface RNAs. The level of RNase 7 in the skin is not quantified to allow an analysis of whether the levels are relevant/responsible for RNA instability on human skin.

The low levels of RNA in other sample types (saliva, urine, sweat, serum) could be due to breakdown in these samples. There is no control of RNA preservation in these commercially sourced samples. The stratum corneum RNA is sampled from 2 healthy volunteers using the method from Benson lab (Wong et al 2004) where circular discs with a diameter of 17 mm are used. Considering that the reported method samples from the entire facial area the comparison to a standard tape strip method would require normalization with regard to sampling area.

2. Does the gene expression of atopic disease markers correlate with the local disease score in the AD patients?

3. How does the transcriptomic profile using this new method correlate with that obtained with standard techniques such as tape stripping and skin biopsies?

4. Is there any difference in the procedure and results of this method when examining lesional vs. non-lesional skin areas?

Reviewer #2 (Remarks to the Author):

The manuscript entitled "Non-invasive human skin transcriptome analysis using mRNA in skin surface lipids" by Inoue et al. describes a novel, non-invasive method to collect mRNA present in human sebum via skin surface lipid sampling on oil blotting films. The manuscript outlines the validation of this methodology and describes an application to determine skin surface lipid transcriptomics in human AD versus healthy skin.

The presented method is an alternative to transcriptomic analysis of inflammatory skin diseases using skin biopsies which is invasive and sometimes not feasible or supported by the patients, particularly when samples are to be taken in the face, or tape stripping which is not overly well suited for transcriptome analysis.

Major comments:

I believe the method described in this paper will be of interest to the research community in immunodermatology, I also believe however that the manuscript is mostly a validation of a novel sampling approach and may be better suited in a journal focussing on novel methods or a clinical dermatology journal. The novel insights into disease understanding of AD are rather limited and one may ask the question what the advantage of this method is over other methods to dissect the pathomechanism of AD, such as single cell RNA seq, spatial transcriptomics, transcriptomics after laser microdissection or separation of dermis and epidermis, because collection of biopsies in AD is usually easily possible.

With regard to disease understanding in AD, I don't think that the selection of the GO terms in figure 6 and 7 are overly helpful as they don't help in understanding possible dysregulation in AD. mRNA splicing, G-Protein coupled receptor signaling, peptide cross linking etc are very general processes. What does it mean that they are up or downregulated in surface Lipid mRNA?

In figure 7 several genes involved in sebocyte differentiation and sebum production are shown, but they are not differentially regulated in AD versus normal skin. Only TGFB is, but the reasoning that this is responsible for reduced sebum production in AD seems far fetched and not substantiated. Scientifically more interesting would have been the analysis of skin surface lipid transcriptomics of diseases like inflammatory acne which is usually difficult to obtain biopsies of, particularly when lesions in the face are of interest. Alternatively, a comparison of the skin surface lipid transcriptome of Acne versus AD could have been of interest given that one disease is characterized by overproduction of sebum, the other by reduced sebum production. In that case novel insights into the different pathomechanism of these diseases with regard to sebocyte biology could have been investigated applying this novel method.

I would therefore see this manuscript rather in a journal focusing on describing new Methods as the novel insights into AD applying this interesting method are rather thin.

Minor comments:

In line 406 the authors talk about 918 upregulated genes and 1033 downregulated genes, but in the next sentence refer to 833 upregulated and 951 down regulated genes. Where does the discrepancy come from? The gene lists should be made available in the supplementary data.

Reviewer #3 (Remarks to the Author):

In this paper the authors present a non-invasive method for assessing mRNA expression in the skin. The method uses skin blotting to isolate mRNA and other biomolecules from the skin surface, with subsequent processing to obtain mRNA. The authors identify differences in AD and healthy cohorts of patients and suggest this method may have widespread utility for understanding skin pathologies and even potentially systemic pathologies.

This is a potentially interesting approach and the benefits of this type of non-invasive profiling for clinical research are significant. The methodology is clearly presented and the initial findings are interesting. However there are some questions raised with respect to the current findings;

1. the study only used 30 male controls/patients between 20-48 yo. Are the findings truly representative - what is the individual variation observed? This would have a significant impact on the utility of this approach if these findings are widely the same across healthy individuals or only relevant to this particular group.

2. Similarly the preparation uses the whole face, which is a large surface area. Is this approach suitable for other areas of skin or only for the face - SG density is much higher on the face than other areas of the body. Again this seems important given the claims that this approach is widely applicable for skin diseases and even potentially systemic disease.
3. The results suggest the mRNAs identified relate largely to the SG (as expected) and do not represent dermis for example as it is not closely linked to SG. Therefore how is this approach likely to be related to systemic pathology and or other skin diseases if the mRNA detected is largely related to some epidermal/hf and primarily SG derived mRNA but not from even fairly proximal tissue? Is there evidence that the SG mRNA would change in other diseases at all?
4. The use of two unrelated plasma/urine samples to measure RNA would have been better to just collect blood/urine from the patients recruited for the study and compared across the fluids collected for individual patients.
5. Other studies have suggested lower TGFb is related to AD, given it has an immunosuppressive effect. In this study TGFb is elevated compared to controls. Therefore is this reflective that the source of RNA only presents a partial picture of the pathology? If this is the case again the proposed wide applicability of this technique is questionable.

AUTHORS' RESPONSE TO REVIEWERS' COMMENTS

Reviewer-1

We appreciate the time and effort you have dedicated to providing insightful feedback on ways to strengthen our paper. We have incorporated the changes that reflect the detailed suggestions that you have graciously provided into the manuscript (red text). We hope that our edits and the responses provided below satisfactorily address all the issues and concerns you have noted.

COMMENTS

1-1. The use of an oil-blotting film and extraction by QIASol indicate that the resulting RNAs are associated with lipids. The transcriptomic analysis further support that they are derived from sebaceous glands.

It is, however, possible that the isolated skin surface RNA could also be derived from other parts of the skin and is not necessarily associated with lipids from the sebaceous glands. A control experiment with a different isolation material (skin wipe) and extraction buffer could be relevant.

Response:

We appreciate your suggestions. We have comprehensively analyzed the RNAs present in SSLs, which include RNAs derived from the epidermis and hair follicles, in addition to those from the sebaceous glands. The reviewer commented that sebaceous gland-derived RNAs are associated with lipids, but RNAs from other parts of the skin are not necessarily associated with lipids; and that RNAs not associated with lipids should be collected using different materials and extraction buffers and analyzed as a control sample. However, it is technically very difficult to separate lipid-associated RNAs and non-associated RNAs from SSL-RNAs, even if the appropriate isolation materials and extraction buffers are available. To make up for this limitation, we investigated the contribution of each source to the total SSL-RNAs by collecting RNAs from each part of the skin with LMD and analyzed the RNA.

In response to the reviewer's comment, as an alternative, we thought that it might be possible to clarify the characteristics of SSL-RNAs by comparing them with stratum corneum RNAs. As an additional experiment, we compared the expression profiles of genes in the SSLs and the stratum corneum in 10 healthy male subjects. Comparisons of the marker genes highly expressed in each skin region revealed that the stratum corneum predominantly expresses epidermal marker genes, whereas SSLs express marker genes of sebaceous gland as well as epidermal marker genes, indicating that SSL-RNAs characteristically contained RNAs derived from sebaceous glands.

We have added a new Figure (Supplementary Fig. 2) and Method (Supplementary Method 2). We have also added relevant descriptions in the Material and Methods (lines 94-98), and the Results

(lines 401-407), and Discussion sections (lines 492-494) concerning this new experiment. We have also modified the description on subject recruitment (lines 74-83).

The revisions are as follows:

Materials and Methods

a) lines 74-83 (revised the sentences)

For SSL-RNA analysis in healthy subjects and in patients with AD, participants were recruited as follows. Thirty-two healthy male individuals (mean age: 34.6 years, range: 20–49 years, SD = 9.24) were recruited in October 2016. The participants were evaluated by dermatologists prior to the commencement of the study to confirm that no obvious skin disease or conditions were present on their faces. Subjects with self-reported AD were recruited, and 30 male patients with mild and moderate facial AD (mean age: 31.0 years, range: 20–48 years, SD = 8.82) diagnosed by the dermatologist between June and October 2017 were then included in the study. To compare SSL-RNAs and stratum corneum RNAs, then healthy male individuals (mean age : 40.6 years, range: 28-57 years, SD = 10.23) were recruited. The participants received an adequate explanation of the study and provided written informed consent.

b) lines 94-98 (added the sentences)

To compare SSL-RNAs and stratum corneum RNAs, SSLs were collected according to the aforementioned method. Then, SSLs were completely removed from the face using facial cleaners, and five stratum corneum samples were collected from each left and right cheek using D-squame tape strips (22 mm in diameter, CuDerm, Dallas, Tex). Samples were stored at -80 °C until use.

Results

a) lines 401-407 (added the sentences)

To further characterize SSL-RNAs, we compared the transcript profiles of SSLs and the stratum corneum. RNAs present in the stratum corneum were enriched in epidermis-related RNAs, whereas RNAs present in the SSL were also rich in sebaceous gland- and hair follicle-related RNAs (Supplementary Fig. 2).

Discussion

a) lines 492-494 (added a sentence)

Overall, in contrast to tape-stripped stratum corneum RNAs, SSL-RNAs were thought to characteristically contain RNAs from the epidermis, along with RNAs from the sebaceous glands and surrounding tissues.

Supplementary Fig. 2

Supplementary Figure 2. mRNA expression in SSLs and stratum corneum.

SSL-RNAs and stratum corneum RNAs were collected from the same 10 healthy male subjects. The expression of genes characteristic of each skin region in SSL-RNAs and stratum corneum RNAs is shown in the heat map.

We have also added the Supplementary Method 2

1-2. The high level of RNA isolated with this method indicate that the RNAs are protected by lipids. However the RNase 7 experiments only provide indirect evidence. It is true that RNase 7 exist in the skin but it is not clear to what extent it protects(degrade?) the surface RNAs. The level of RNase 7 in he skin is not quantified to allow an analysis of whether the levels are relevant/responsible for RNA instability on human skin.

Response :

We performed our experiments using RNase 7 because comprehensive analysis of SSL-proteins—collected from 3 healthy subjects by LC-MS/MS—revealed that only RNase 7 was detected among the RNases. In response to the reviewer’s comment, we performed additional

experiments to reveal the RNase 7 content in SSL. The median concentration of RNase 7 present in SSL collected from 11 subjects was 0.09 ng/mg-sebum (range: 0.026–0.38 ng/mg). The results of our *in vitro* experiments were more severe than this value would indicate, therefore, we believe that RNase 7 can contribute to RNA degradation *in vivo*, and that this action can be inhibited by sebum lipids. This information has been added in the Materials and Methods (lines 176–181), and Results (lines 329–330) sections.

The revisions are as follows:

Materials and Methods

- a) lines 176–181 (added the sentences)

Enzyme-linked immunosorbent assay for RNase 7

After the facial sebum collected from 11 healthy subjects was weighed, total protein was extracted using 400 μ L of RIPA buffer supplemented with protease/phosphatase inhibitor (Cell Signaling Technology, Danvers, MA, USA). The concentration of RNase 7 was measured using an RNase 7 ELISA kit (Hycult Biotech, Uden, Netherlands) according to the manufacturer's protocol.

Results

- a) lines 329–330 (added a sentence)

The median RNase 7 content in SSLs collected from 11 subjects was 0.09 ng/mg-sebum (range: 0.026–0.38 ng/mg).

1-3. The low levels of RNA in other sample types (saliva, urine, sweat, serum) could be due to break down in these samples. There is no control of RNA preservation in these commercially sourced samples.

Response:

Although the instructions attached to the product state that the samples were stored at -80 °C immediately after collection, the possibility of RNA degradation cannot be completely ruled out. Meanwhile, in another comment, you pointed out the problem associated with the of standardization of different types of samples. We are currently conducting a comparative analysis of the expression of *ACTB* and *GAPDH* using RNA from the same volume of samples, but the gene expression read-outs in liquid samples may increase if the volume used for RNA extraction is increased. In addition, it is difficult to compare RNA expression levels in SSLs collected using oil-blotting film with those in liquid samples, such as urine, sweat, and serum. Furthermore, the results of gene expression analyses between SSLs and other samples in the absence of standardized comparisons are likely to be misleading. Even if we conduct new analyses with fresh samples, we would not be able to solve the aforementioned problems. In Fig. 1a, we considered it more important to demonstrate

that human mRNAs were contained in SSLs than to compare the expression levels of *ACTB* and *GAPDH* between SSLs and other biological samples. Therefore, we have deleted the data on urine, sweat, serum, saliva and stratum corneum from Fig. 1a of the original manuscript. We have also revised the Methods (deleted lines 101-105 in the original manuscript) and Results (lines 308-310) sections.

The revisions are as follows:

Method

a) lines 101-105 of original manuscript (deleted the sentences)

Human saliva, sweat, urine, and serum samples collected from two donors were purchased from Cosmo Bio (Tokyo, Japan). Total RNA was extracted from 1 mL of each sample using the TRIzol LS reagent (Thermo Fisher Scientific, Waltham, MA, USA). Total RNA was extracted from the stratum corneum of the cheek of two healthy males according to a previous report (12).

Results

a) lines 308-310 (revised the sentences)

qPCR was conducted to evaluate the mRNA abundance in the SSL samples. The expression of *ACTB* and *GAPDH* mRNA in SSLs was comparable that in 100 ng–500 pg and 1 ng–100 ng of total RNA in NHEK, respectively (Fig. 1a).

Figure 1a (revised)

Deleted data on urine, sweat, serum, saliva and stratum corneum from Fig. 1a of the original manuscript. New Figure 1a is as follows;

1-4. The stratum corneum RNA is sampled from 2 healthy volunteers using the method from Benson lab (Wong et al 2004) where circular discs with a diameter of 17 mm are used.

Considering that the reported method samples from the entire facial area the comparison to a standard tape strip method would require normalization with regard to sampling area.

Response :

As mentioned in comments 1–3, we have removed the data comparing gene expression in SSLs and stratum corneum from Fig. 1a.

2. Dose the gene expression of atopic disease markers correlate with the local disease score in the AD patients?

Response :

In this study, dermatologists diagnosed the local AD severity in faces according to the guidelines of the Japanese Dermatological Association. Six patients with mild disease and 24 patients with moderate disease were included in this study. Although we verified the relationship between the severity of AD and the expression of genes shown in Fig. 5a, no significant differences were noted between the mild and moderate patient groups because of the insufficient number of mild patients. To clarify the relationship with local disease score, we believe that larger validation studies are needed in the future. We have mentioned this point in the Discussion section as a limitation (lines 535-539).

The revisions are as follows:

Discussion

a) lines 535-539 (added the sentences)

There is one limitation regarding the clinical findings of the SSL-RNA analysis. In this study, we validated this method in a relatively limited number of patients (samples collected from 29 healthy male subjects and 30 patients with AD aged 20-48 years were used to validate this method). Validation in a more subjects and on a larger scale is needed to further strengthen the findings of this study and to confirm the applicability of the SSL-RNA analysis method.

3. How does the transcriptomic profile using this new method correlated with that obtained with standard techniques such as tape stripping and skin biopsies?

Response :

To answer this question, we conducted an additional experiment (Supplementary Fig. 2). Skin biopsy samples contain RNA information from all regions of the skin, including epidermis, sebaceous glands, sweat glands, hair follicles, and dermis. In contrast, as shown in this study, the analysis of stratum corneum RNA provided expressional information mainly in the epidermis, while

the analysis of SSL-RNA provided expression-related information not only in the epidermis, but also in the sebaceous glands and hair follicles.

The correlation coefficient for RNAs detected in both SSL-RNAs and stratum corneum RNAs was 0.51 (average value of 10 subjects). When focusing on 34 genes highly expressed in the epidermis, the correlation coefficient increased to 0.82. These results suggest that the commonality of RNA origin in skin tissues has an effect on the correlation of transcriptome profiles, and that SSL-RNAs reflect the gene expression status of the skin tissues.

Unfortunately, we were not able to obtain biopsy samples of the face; therefore, a detailed comparison that includes those samples as well will be a future investigation.

We have added this information to the Results (lines 401-407) and Discussion (lines 492-494) sections. We have added a new Figure (Supplemental Fig. 2). Please also see COMMENTS-1-1.

The revisions are as follows:

Results

a) lines 401-407 (added the sentences)

To further characterize SSL-RNAs, we compared the transcript profiles of SSLs and the stratum corneum. RNAs present in the stratum corneum were enriched in epidermis-related RNAs, whereas RNAs present in the SSL were also rich in sebaceous gland- and hair follicle-related RNAs (Supplementary Fig. 2). The correlation coefficient between SSL-RNAs and stratum corneum RNAs calculated using transcriptome profile normalized with DESeq2 was 0.51 (the mean value of 10 subjects). When focusing on 34 genes highly expressed in the epidermis, the correlation coefficient increased to 0.82.

Discussion

a) lines 492-494 (added a sentence)

Overall, in contrast to tape-stripped stratum corneum RNAs, SSL-RNAs were thought to characteristically contain RNAs from the epidermis, along with RNAs from the sebaceous glands and surrounding tissues.

4. Is there any difference in the procedure and results of this method when examining lesional vs. non-lesional skin areas?

Response :

As SSLs were not collected from the non-lesional skin of patients with AD, we cannot show the differences between lesional and non-lesional skin in this study. However, SSLs can be collected from non-lesional skin using the same procedure (wiping using oil-blotting film). In our preliminary study, we confirmed that the SSL-RNA profile is different between the lesional and non-lesional skin of AD patients.

Other revisions:

The entire manuscript has been reviewed and some corrections have been made.

AUTHORS' RESPONSE TO REVIEWERS' COMMENTS

Reviewer-2

We appreciate the time and effort you have dedicated to providing insightful feedback on ways to strengthen our paper. We have incorporated the changes that reflect the detailed suggestions you have graciously provided into the manuscript in red text. We hope that our edits and the responses provided below satisfactorily address all the issues and concerns you have noted.

COMMENTS

Major comments:

I believe the method described in this paper will be of interest to the research community in immunodermatology, I also believe however that the manuscript is mostly a validation of a novel sampling approach and may be better suited in a journal focussing on novel methods or a clinical dermatology journal. The novel insights into disease understanding of AD are rather limited and one may ask the question what the advantage of this method is over other methods to dissect the pathomechanism of AD, such as single cell RNA seq, spatial transcriptomics, transcriptomics after laser microdissection or separation of dermis and epidermis, because collection of biopsies in AD is usually easily possible.

With regard to disease understanding in AD, I dont think that the selection of the go Terms in figure 6 and 7 are overly helpful as they dont help in understanding possible dysregulation in AD. mRNA splicing, G-Protein coupled receptor signaling, peptide cross linking etc are very general processes. What does it mean that they are up or downregulated in surface Lipid mRNA?

Response :

As you mentioned above, the AD skin tissue can be collected by biopsy and analyzed by various methods. However, in a clinical setting, it is not practical to repeatedly obtain facial skin samples from patients by punch biopsy. The clinical phenotypes of AD are complex, and AD recurs with exacerbations and remissions; therefore, it is difficult to understand the complex pathology of AD with a limited number of samples and a snapshot of a single point in time. As our method uses non-invasive and easily collectable SSL-RNAs, we expect that it will contribute to understanding the pathophysiology of each patient with AD, and to the planning of personalized treatment. We have added a sentence to the Discussion section (lines 547-549).

DAVID analysis revealed that only limited GO terms were enriched in AD patients, in contrast to the results of individual molecular expression analysis shown in Figure 5. We conducted a literature review on the association between AD pathology and the functions of the 32 genes that make up the GO term for mRNA splicing (GO:0000398, mRNA splicing, via spliceosome). However, we could not obtain any clear correlations between them. With respect to downregulated genes, "GO:0007186

G-protein coupled receptor signaling pathway” and “GO:0018149 peptide cross-linking” had many overlapping genes that formed “GO:0050907 detection of chemical stimulus involved in sensory perception” and “GO:0031424 keratinization”, respectively. These GO terms (GO:0007186 and GO:0050907) were also identified in individuals with sensitive skin (data not shown) in our other research project. Since sensitive skin is a symptom often observed in patients with AD, we speculate that SSL-RNAs reflect one aspect of AD pathology, and that these pathways identified as GO terms may contribute, at least in part, to the pathogenesis of AD. The relationship between these pathways identified as GO terms and the pathogenesis of AD will, of course, need to be studied in detail in the future. In addition, we have modified a sentence (lines 517-521).

The revisions are as follows:

Discussion

a) lines 547-549 (added a sentence)

This comprehensive analysis method using SSL-RNAs is expected to contribute to solving these clinical and research issues owing to advantageous features such as non-invasiveness and ease of sampling.

b) lines 517-521 (revised the sentence)

“GO:0050911: detection of chemical stimulus involved in sensory perception of smell” was the most significantly changed GO term in our study, consistent with the analysis of stratum corneum RNAs collected from patients with AD (13), indicating that the analysis of SSL-RNAs successfully captured the characteristics of AD.

In figure 7 several genes involved in sebocyte differentiation and sebum production are shown, but they are not differentially regulated in AD versus normal skin. Only *TGFβ* is, but the reasoning that this is responsible for reduced sebum production in AD seems far fetched and not substantiated.

Response :

As the reviewers have pointed out, it has not been proven that increased *TGFβ* expression actually contributes to reduced sebum production in patients with AD. However, we believe that our results suggest this possibility. Moreover, as the analysis of the mechanism underlying sebum production in AD patients is not the main theme of this study, such a detailed analysis would fit better as our future investigation.

In response to your comment, we have added a sentence to the Discussion section (lines 531-532).

The revisions are as follows:

Discussion

a) lines 531-532 (added a sentence)

The possibility of TGF- β -mediated regulation of sebum production requires further investigation.

Scientifically more interesting would have been the analysis of skin surface lipid transcriptomics of diseases like inflammatory acne which is usually difficult to obtain biopsies of, particularly when lesions in the face are of interest. Alternatively, a comparison of the skin surface lipid transcriptome of Acne versus AD could have been of interest given that one disease is characterized by overproduction of sebum, the other by reduced sebum production. In that case novel insights into the different pathomechanism of these diseases with regard to sebocyte biology could have been investigated applying this novel method.

Response :

Thank you very much for your valuable advice. SSL-RNA analysis is indeed a promising tool, especially in the assessment of skin disease with facial involvement. We would like to use this method to analyze various skin diseases, including acne, in the future.

Minor comments:

In line 406 the authors talk about 918 upregulated genes and 1033 downregulated genes, but in the next sentence refer to 833 upregulated and 951 down regulated genes. Where does the discrepancy come from? The gene lists should be made available in the supplementary data.

Response :

Thank you for pointing this out. We used the Entrez gene IDs as sheet output from the Ion Torrent Suite Software Plugins to perform GO analysis. However, the number of genes used in GO analysis was 833 and 951, because of incomplete linkage between the Gene symbols and the Entrez gene IDs in the sheet. We retrieved all the missing Entrez gene IDs and reanalyzed them against all 918 upregulated genes and 1033 downregulated genes. We have revised the Results section (lines 434 and 437) accordingly. In the result of the reanalysis, the extracted GO terms were not changed, but the FDR values were slightly changed. Fig. 6c was replaced accordingly. Furthermore, we have submitted the gene list as supplemental data.

The revisions are as follows:

Results

a) lines 434 and 437 (revised a sentence)

For the 918 upregulated genes, the GO terms “mRNA splicing” and “stimulatory C-type lectin receptor signaling pathway” were significantly enriched, while the GO term “detection of chemical stimulus involved in sensory perception of smell and keratinocyte differentiation” was enriched among the 1033 downregulated genes (Fig. 6c).

Other revisions:

The entire manuscript has been reviewed and some corrections made.

AUTHORS' RESPONSE TO REVIEWERS' COMMENTS

Reviewer-3

We appreciate the time and effort you have dedicated to providing insightful feedback on ways to strengthen our paper. We have incorporated the changes that reflect the detailed suggestions you have graciously provided in the manuscript in red text. We also hope that our edits and the responses provided below satisfactorily address all the issues and concerns you have noted.

COMMENTS

1. the study only used 30 male controls/patients between 20-48 yo. Are the findings truly representative - what is the individual variation observed? This would have a significant impact on the utility of this approach if these findings are widely the same across healthy individuals or only relevant to this particular group.

Response :

In this study, we performed SSL-RNA analysis on male subjects who were clinically diagnosed with AD and found that healthy subjects and patients with AD showed characteristic molecular profiles. The clinical phenotypes of AD are complex, and there are marked differences between individuals. As shown in Figures 4 and 5, the RNA expression profiles of patients with AD vary between individuals, but the overall expression of RNAs that are closely related to AD is consistent with previous reports. Due to the bias of the subject population in this study, we cannot say definitively whether the results are representative or not; however, we believe that we have captured at least a part of the typical phenomena of AD, since we can confirm similarities with previous reports. In response to your comment, we have described the limitation regarding the bias of the subject population in the Discussion section (lines 535-539).

The revisions are as follows:

Discussion

a) lines 535-539 (added the sentences)

There is one limitation regarding the clinical findings of the SSL-RNA analysis. In this study, we validated this method in a relatively limited number of patients (samples collected from 29 healthy male subjects and 30 patients with AD aged 20-48 years were used to validate this method). Validation in a more subjects and on a larger scale is needed to further strengthen the findings of this study and to confirm the applicability of the SSL-RNA analysis method.

2. Similarly the preparation uses the whole face, which is a large surface area. Is this approach suitable for other areas of skin or only for the face - SG density is much higher on

the face than other areas of the body. Again this seems important given the claims that this approach is widely applicable for skin diseases and even potentially systemic disease.

Response :

We have confirmed that SSLs can also be collected from the scalp, chest, back, and buttocks, and can be analyzed in the same way. As the reviewer commented, there is a correlation between the density of sebaceous glands and the amount of SSL-RNAs that can be collected. We would like to refrain from further mentioning these findings because we are preparing to publish them in another paper. However, in response your comment, we have added a “Note” in the Supplemental Methods (bottom line). In regard to the potentially systemic disease, please also refer to the Author’s Response to Comment-3.

The revisions are as follows:

Supplementary Methods 1

a) Bottom line (added a sentence)

Note: SSL-RNAs can be collected and analyzed from scalp, chest, back, and buttocks.

3. The results suggest the mRNAs identified relate largely to the SG (as expected) and do not represent dermis for example as it is not closely linked to SG. Therefore how is this approach likely to be related to systemic pathology and or other skin diseases if the mRNA detected is largely related to some epidermal/hf and primarily SG derived mRNA but not from even fairly proximal tissue? Is there evidence that the SG mRNA would change in other diseases at all?

Response :

We are now investigating the relationship between systemic pathologies and expression of SSL-RNAs. So far, we have found that at least some neurodegenerative, metabolic, and inflammatory diseases affect SSL-RNA expression and that these diseases can be distinguished based on SSL-RNA expression profiles (Unpublished data). These systemic diseases have been reported to show some phenotype in the human skin. Therefore, we consider that SSL-RNAs have the potential to be applied not only to local skin but also to systemic pathophysiological analysis. However, we would like to avoid mentioning these points in this paper because we are presenting these findings in other papers.

4. The use of two unrelated plasma/urine samples to measure RNA would have been better to just collect blood/urine from the patients recruited for the study and compared across the fluids collected for individual patients.

Response :

As pointed out by you, although the instructions attached to the product state that the samples were stored at -80 °C immediately after collection, we cannot rule out the possibility of RNA degradation. Moreover, we noticed that there is a problem of normalization of different types of samples. It is difficult to compare RNA expression levels in SSLs collected using oil-blotting film with those in liquid samples, such as urine, sweat, and serum. Furthermore, the results of gene expression analyses between SSLs and other samples in the absence of standardized comparisons are likely to be misleading. Even if we conduct new analyses with fresh samples, we would not be able to solve the aforementioned problems. In Fig. 1a, we considered it more important to demonstrate that human mRNAs were contained in SSLs than to compare the expression levels of *ACTB* and *GAPDH* between SSLs and other biological samples. Therefore, we have deleted the data on urine, sweat, serum, saliva and stratum corneum from Fig. 1a of the original manuscript. We have also revised sentences in the Methods (deleted a sentence in the original manuscript, lines 101-105) and Results (deleted a paragraph in the original manuscript, lines 288–291 and revised line 308-310 in the revised manuscript) sections and have modified a Figure (deleted a data of *ACTB* and *GAPDH* in urine, sweat, serum, saliva and stratum corneum from Fig. 1a of the original manuscript).

The deleted parts are as follows:

Method

a) lines 101-105 of original manuscript (deleted the sentences)

Human saliva, sweat, urine, and serum samples collected from two donors were purchased from Cosmo Bio (Tokyo, Japan). Total RNA was extracted from 1 mL of each sample using the TRIzol LS reagent (Thermo Fisher Scientific, Waltham, MA, USA). Total RNA was extracted from the stratum corneum of the cheek of two healthy males according to a previous report (12).

Results

a) lines 308-310 (revised the sentences)

qPCR was conducted to evaluate the mRNA abundance in the SSL samples. The expression of *ACTB* and *GAPDH* mRNA in SSLs was comparable that in 100 ng–500 pg and 1 ng–100 ng of total RNA in NHEK, respectively (Fig. 1a).

Figure 1a

Deleted a data of urine, sweat, serum, saliva and stratum corneum from Fig. 1a of the original manuscript. New Figure 1a is as follows;

5. Other studies have suggested lower TGF β is related to AD, given it has an immunosuppressive effect. In this study TGF β is elevated compared to controls. Therefore is this reflective that the source of RNA only presents a partial picture of the pathology? If this is the case again the proposed wide applicability of this technique is questionable.

Response:

Yes, some reports indicated that the expression of *TGF β* RNA in PBMCs from AD patients was significantly decreased compared with that in healthy subjects (Ann Allergy Asthma Immunol. 2000, 84:553-558). On the other hand, the expression of *TGF β* in human skin tissues collected from patients with AD by punch biopsy has been reported to be significantly elevated compared with that in healthy subjects (J Allergy Clin Immunol. 2003, 111:875-81). Considering that SSL-RNAs mainly reflect gene expression in the skin, we believe that our findings are not necessarily inconsistent with previous reports. In response to your comment, we have added relevant sentences in the Discussion section (lines 528-529, 531-532).

The revisions are as follows:

Discussion

a) lines 528-529 (added a sentence)

Increase in the number of TGF- β -positive cells in patients with AD has also been reported (55).

b) lines 531-532 (added a sentence)

The possibility of TGF- β -mediated regulation of sebum production requires further investigation.

Other revisions:

The entire manuscript has been reviewed and some corrections have been made.

Reviewers' comments:

Reviewer #1 (Remarks to the Author):

As far as I can see the only one of my comments that was addressed was the concentration of RNase7 on the skin. It was determined to be 0.026–0.38 ng/mg. It should be discussed that this level is well below the concentration used in the RNase7 stability assay (30-1000 ng/mL).

I still think the manuscript is relevant and interesting and support publication

Reviewer #2 (Remarks to the Author):

There are a number of open questions how representative the method is and how it compares to more established Methods of collecting RNA from the skin. New insights into the biology of AD are very limited and it cannot even be excluded that they may be misleading by sampling RNA in this way. A comparison of diseases with high versus low sebum production versus healthy skin or a more in depth comparison of this way of RNA sampling with established ways of sampling may have therefore been a more thorough approach. As it stands at the moment is remains unclear if this method is really suitable to generate new insights about AD or whether this sampling method even leads to misleading results.

Reviewer #3 (Remarks to the Author):

The authors have responded to many comments and the manuscript is significantly improved. However the key purpose of the paper appears to be to provide evidence that this methodology has potential for the understanding of skin diseases overall. This is evidenced by the abstract first and concluding sentences "Non-invasive acquisition of mRNA data from the skin can be extremely useful for understanding skin physiology and diseases..... These results indicate that the analysis of SSL-RNAs is a promising strategy to understand the pathophysiology of skin diseases." Without the inclusion of other body site data, a larger population base (including female/age groups) or work demonstrating changes associated with other skin diseases, I do not think the authors have really validated this approach as being useful for this purpose. Rather this manuscript provides some data suggesting this approach provides some findings that align with what is already known about AD, and others that are conflicting. From the author responses it is clear that other data exists but they are, understandably, keen to present this in other manuscripts. However this does leave the current manuscript with limited impact given the issues highlighted previously.

AUTHORS' RESPONSE TO REVIEWERS' COMMENTS

Reviewer-1

We appreciate the time and effort you have dedicated to providing insightful feedback on ways to strengthen our paper. We have incorporated the changes that reflect the suggestions that you have graciously provided into the manuscript (red text). We hope that our edits and the responses provided below satisfactorily address all the issues and concerns you have noted.

COMMENTS

As far as I can see the only one of my comments that was addressed was the concentration of RNase7 on the skin. It was determined to be 0.026–0.38 ng/mg. It should be discussed that this level is well below the concentration used in the RNase7 stability assay (30-1000 ng/mL).

Response :

The *in vitro* evaluation showed that RNase7 degrades RNA (Fig. 1c/d, Fig. 2c). In addition, the results of our measurements confirmed that skin surface lipids (SSLs) contain an average of 0.09 ng per mg of sebum of RNase7 (Lines 340–341). Furthermore, as mentioned in our previous response letter, RNase7 is considered to be the major RNase detected in sebum. From these results, it is inferred that RNase7 can contribute to RNA degradation. Strictly speaking, however, it is difficult to compare the concentrations of RNase 7 in the same way because the evaluation of RNA degradation *in vitro* is performed in an aqueous solution system (30-1000 ng/mL), and the RNase7 in SSLs is calculated as the amount per weight of sebum. Therefore, in this study, we believe it is appropriate to mention only the fact that RNase7 exists in sebum (Lines 340–341) and that RNase7 can degrade RNA (Fig. 1c/d, Fig. 2c), and to refrain from having a quantitative discussion of them.

AUTHORS' RESPONSE TO REVIEWERS' COMMENTS

Reviewer-2

We appreciate the time and effort you have dedicated to providing insightful feedback on ways to strengthen our paper. We have incorporated the changes that reflect the suggestions you have graciously provided into the manuscript in red text. We hope that our edits and the responses provided below satisfactorily address all the issues and concerns you have noted.

COMMENTS

There are a number of open questions how representative the method is and how it compares to more established Methods of collecting RNA from the skin.

New insights into the biology of AD are very limited and it cannot even be excluded that they may be misleading by sampling RNA in this way.

A comparison of diseases with high versus low sebum production versus healthy skin or a more in depth comparison of this way of RNA sampling with established ways of sampling may have therefore been a more thorough approach. As it stands at the moment is remains unclear if this method is really suitable to generate new insights about AD or whether this sampling method even leads to misleading results.

Response :

As you pointed out, it is expected in all cases that the results will vary depending on the sampling method and analysis. Therefore, we clarified which region/cell population information in the skin can be mainly analyzed by this method by performing a detailed analysis with laser microdissection (LMD) (Figure 4). In addition, a comparative analysis using SSL-RNAs and stratum corneum RNAs was performed to show the differences between our method and existing methods (Supplementary Figure 2). You may have asked for a comparison with the skin biopsy method as an established method, but it could not be done because biopsy of facial skin was difficult in terms of invasiveness and ethics. Although we were not able to compare our results with all existing methods, we believe that our findings on the origin of SSL-RNAs presented in this study will provide certain information necessary to understand the results of the analysis by this method. You also note that the SSL-RNA results may be misleading, but we believe that this concern can be considerably alleviated by understanding the origin of the SSL-RNAs. In fact, as we have verified in this study, the findings obtained by SSL-RNA analysis do not contradict the findings obtained by existing methods. Since our human study was conducted in subjects with mild to moderate atopic dermatitis symptoms, we may not have discovered anything new. We would like to examine this point in patients with more severe AD in the future.

According to your suggestion, we performed a comparative analysis of SSL-RNA expression in male and female subjects known to have different sebum production. This study was conducted with

38 healthy male subjects (mean age: 39.1 years, range: 20–58 years, SD = 12.5) and 42 healthy female subjects (mean age: 38.5 years, range: 20–56 years, SD = 11.6). The SSL-RNA analysis revealed that the expression levels of the genes differed greatly depending on skin characteristics; the expression levels of 25 lipid metabolism-related genes expressed in sebaceous glands were higher in male individuals in agreement with the recovered sebum levels (Supplementary Figure 4). Based on these findings, we believe that SSL-RNAs can reflect not only skin pathology such as atopic dermatitis, but also differences in skin conditions in healthy individuals.

We have therefore added the information to the revised manuscript as follows:

Method

a) lines 118–121 (added sentences)

Measurement of sebum secretion.

Facial sebum levels on the forehead and cheek were measured using a Sebumeter (SM815, C-K Electronics, Cologne, Germany). The casual sebum levels before washing the face and the recovered sebum levels during 60 minutes after washing the face were measured (19).

Results

a) lines 431–437 (added sentences)

Furthermore, we performed a comparative analysis of SSL-RNA profiles in male and female individuals known to have different sebum production levels (50). The levels of casual sebum on the cheek and the recovered sebum on the forehead and cheek at 60 minutes after washing the face in males were significantly higher than those in female (Supplementary Fig. 4a). The expression of the SSL-RNAs differed greatly depending on such skin characteristics; the expression levels of 25 lipid metabolism-related genes expressed in sebaceous glands were higher in males, in agreement with recovered sebum levels (Supplementary Figure 4b).

Discussion

a) lines 542–546 (added sentences)

Finally, we verified the applicability of this method by performing a comparative analysis of the SSL-RNA profiles of healthy subjects with different amounts of sebum production, and healthy subjects and patients with AD. The SSL-RNA analysis revealed that a group of genes involved in lipid synthesis is highly expressed in sebaceous glands in men with high amounts of recovered sebum, indicating that this method may be useful for detecting skin conditions.

b) lines 573–576 (modified a sentence)

This non-invasive method has potential application in unraveling the molecular profiles of skin conditions and diseases including AD, which will be helpful for clinical management of these diseases in the future.

Supplementary Figure 4

Supplementary Figure 4. Comparison the casual and the recovered sebum levels between males and females.

Facial sebum secretions (forehead and cheek) were measured using a Sebumeter. (a) The casual and recovered sebum levels in forehead and cheek. The level of significance of differences among the groups was analyzed using Student *t*-test. * $p < 0.05$, ** $p < 0.01$. (b) Heatmaps using z-transformed \log_2 (normalized counts + 1) on 25 lipid metabolism-related genes that were highly expressed in sebaceous glands (Fig. 7b). N.S.: not significant. Females ($n = 42$), males ($n = 38$).

Other revisions:

The entire manuscript has been reviewed and some corrections have been made.

AUTHORS' RESPONSE TO REVIEWERS' COMMENTS

Reviewer-3

We appreciate the time and effort you have dedicated to providing insightful feedback on ways to strengthen our paper. We have incorporated the changes that reflect the suggestions you have graciously provided in the manuscript in red text. We also hope that our edits and the responses provided below satisfactorily address all the issues and concerns you have noted.

COMMENTS

The authors have responded to many comments and the manuscript is significantly improved. However the key purpose of the paper appears to be to provide evidence that this methodology has potential for the understanding of skin diseases overall. This is evidenced by the abstract first and concluding sentences “Non-invasive acquisition of mRNA data from the skin can be extremely useful for understanding skin physiology and diseases..... These results indicate that the analysis of SSL-RNAs is a promising strategy to understand the pathophysiology of skin diseases.”

Without the inclusion of other body site data, a larger population base (including female/age groups) or work demonstrating changes associated with other skin diseases, I do not think the authors have really validated this approach as being useful for this purpose. Rather this manuscript provides some data suggesting this approach provides some findings that align with what is already known about AD, and others that are conflicting. From the author responses it is clear that other data exists but they are, understandably, keen to present this in other manuscripts. However this does leave the current manuscript with limited impact given the issues highlighted previously.

Response:

In response to your comment, we have added the following new information for application of the SSL-RNA method for the analysis of other skin conditions of a larger population base (including male and female subjects with a wide age range).

1. Comparative analysis of SSL-RNAs in different skin regions

To verify the use of sebum collected from areas other than the face as well as the usefulness of this method, we compared SSL-RNAs profiles in the face and scalp from the same subjects. The expression of hair keratin genes was markedly different between the face and scalp, indicating that the SSL-RNA analysis can characterize skin regions (Supplementary Figure 3, Supplementary Table 1).

2. Application of SSL-RNA method to the analysis of other skin conditions

As an application of the SSL-RNA analysis method, we performed a comparative analysis of SSL-RNA profiles in male and female subjects known to have different sebum production levels. This study was conducted with 38 healthy male subjects (mean age: 39.1 years, range: 20–58 years, SD = 12.5) and 42 healthy female subjects (mean age: 38.5 years, range: 20–56 years, SD = 11.6). The SSL-RNA analysis revealed that the expression levels of the genes differed greatly depending on skin characteristics; the expression levels of 25 lipid metabolism-related genes expressed in sebaceous glands were higher in male individuals in agreement with the recovered sebum levels (Supplementary Figure 4). Based on these findings, we believe that SSL-RNAs can reflect not only skin pathology such as atopic dermatitis, but also differences in skin conditions in healthy individuals.

We have therefore added the information to the revised manuscript as follows:

Method

a) lines 83–88 (added sentences)

Furthermore, to evaluate the differences of SSL-RNA profile between face and scalp, 10 healthy female individuals (mean age: 26.3 years, range: 24–32 years, SD = 2.41) were recruited. To analyze the relationship between sebum secretion and SSL-RNAs profile, 38 healthy male individuals (mean age: 39.1 years, range: 20–58 years, SD = 12.5) and 42 healthy female individual (mean age: 38.5 years, range: 20–56 years, SD = 11.6) were recruited.

b) lines 119–121 (added sentences)

Measurement of sebum secretion.

Facial sebum levels on the forehead and cheek were measured using a Sebumeter (SM815, C-K Electronics, Cologne, Germany). The casual sebum levels before washing the face and the recovered sebum levels during 60 minutes after washing the face were measured (19).

Results

a) lines 419–430 (added sentences)

The SSL-RNA profiles of the face and scalp, which have dense sebaceous glands and hair follicles, were also compared. The analysis of the dimensionality reduction using t-distributed stochastic neighbor embedding (t-SNE) and variance stabilizing transformation (VST) values in all genes showed that face samples and scalp samples showed to be distinctly classified into two groups (Supplementary Fig. 3). The expression levels of 633 genes were significantly different between the face and scalp, and gene ontology analysis of 305 genes, which were abundantly expressed in the scalp compared with the face, demonstrated the GO terms “keratinization (GO:0031424), (FDR = 2.43E-10)”. Therefore, the expression levels of hair keratins and hair follicle-specific epithelial keratins (40) were compared between the face and scalp, showing that

the expression levels of most keratin molecules were higher in the scalp than in the face (Supplementary Table 1). Thus, the SSL-RNA analysis can be used to characterize skin regions.

b) lines 431–437 (added sentences)

Furthermore, we performed a comparative analysis of SSL-RNA profiles in male and female individuals known to have different sebum production levels (50). The levels of casual sebum on the cheek and the recovered sebum on the forehead and cheek at 60 minutes after washing the face in male were significantly higher than those in female (Supplementary Fig. 4a). The expression of the SSL-RNAs differed greatly depending on such skin characteristics; the expression levels of 25 lipid metabolism-related genes expressed in sebaceous glands were higher in males, in agreement with recovered sebum levels (Supplementary Figure 4).

Discussion

a) lines 542–546 (added sentences)

Finally, we verified the applicability of this method by performing a comparative analysis of the SSL-RNA profiles of healthy subjects with different amounts of sebum production, and healthy subjects and patients with AD. The SSL-RNA analysis revealed that a group of genes involved in lipid synthesis is highly expressed in sebaceous glands in men with high amounts of recovered sebum, indicating that this method may be useful for detecting skin conditions.

b) lines 573–576 (modified a sentence)

This non-invasive method has potential application in unraveling the molecular profiles of skin conditions and diseases including AD, which will be helpful for clinical management of these diseases in the future.

Supplementary Figure 3

Supplementary Figure 3. Characterization of SSL-RNAs profiles in face and scalp.
t-SNE analysis using variance stabilizing transformation (VST) values for all genes (red, face; green, scalp).

Supplementary Table 1

Supplementary Table 1. The expressions of hair keratins and hair follicle-specific epithelial keratins between face and scalp.

Hair keratin (Type 2)

	Mean of \log_2 (normalized counts) + 1		FC (scalp / face)	FDR
	Face	Scalp		
KRT81	4.74	7.84	1.66	0.321
KRT82	1.06	6.40	6.02	3.54E-08
KRT83	2.97	6.29	2.12	0.429
KRT84	1.00	9.19	9.19	1.41E-16
KRT85	1.87	8.06	4.32	0.045
KRT86	5.91	5.18	0.88	0.880

Hair follicle-specific epithelial keratins (Type 1)

	Mean of \log_2 (normalized counts) + 1		FC (scalp / face)	FDR
	Face	Scalp		
KRT25	9.59	17.20	1.79	5.69E-16
KRT26	3.83	7.66	2.00	0.00777
KRT27	11.54	18.55	1.61	9.10E-39
KRT28	2.19	6.17	2.82	0.306

Hair follicle-specific epithelial keratins (Type 2)

	Mean of \log_2 (normalized counts) + 1		FC (scalp / face)	FDR
	Face	Scalp		
KRT71	11.50	17.75	1.54	2.63E-35
KRT72	9.50	16.55	1.74	1.94E-27
KRT73	1.00	5.92	5.92	3.50E-09
KRT74	4.67	13.34	2.85	4.78E-09
KRT75	4.00	9.59	2.40	0.353

FDR: Benjamini-Hochberg adjusted p -values are shown from the likelihood ratio test between face and scalp. Face ($n = 10$), scalp ($n = 10$). FC: fold change.

Supplementary Figure 4

Supplementary Figure 4. Comparison the casual and the recovered sebum levels between males and females.

Facial sebum secretions (forehead and cheek) were measured using a Sebumeter. (a) The casual and recovered sebum levels in forehead and cheek. The level of significance of differences among the groups was analyzed using Student *t*-test. * $p < 0.05$, ** $p < 0.01$. (b) Heatmaps using z-transformed log₂ (normalized counts + 1) on 25 lipid metabolism-related genes that were highly expressed in sebaceous glands (Fig. 7b). N.S.: not significant. Females (n = 42), males (n = 38).

Other revisions:

The entire manuscript has been reviewed and some corrections have been made.